# Exposing Vulnerabilities in Explanation for Time Series Classifiers via Dual-Target Attacks

Bohan Wang[1]  Zewen Liu[1]  Lu Lin[2]  Hui Liu[3]  Li Xiong[1]  Ming Jin[4]  Wei Jin[1]

## Abstract

Interpretable time series deep learning systems are often assessed by checking temporal consistency on explanations, implicitly treating this as evidence of robustness. We show that this assumption can fail: Predictions and explanations can be adversarially decoupled, enabling targeted misclassification while the explanation remains plausible and consistent with a chosen reference rationale. We propose TSEF (Time Series Explanation Fooler), a dual-target attack that jointly manipulates the classifier and explainer outputs. In contrast to single-objective misclassification attacks that disrupt explanation and spread attribution mass broadly, TSEF achieves targeted prediction changes while keeping explanations consistent with the reference. Across multiple datasets and explainer backbones, our results consistently reveal that explanation stability is a misleading proxy for decision robustness and motivate coupling-aware robustness evaluations for trustworthy time series tasks.

## 1. Introduction

Deep neural networks (DNNs) have achieved success in time series classification, enabling breakthroughs in healthcare, finance, and industrial systems (Liu et al., 2025a; Wen et al., 2024; Zhou et al., 2023; Mohammadi Foumani et al., 2024; Wang et al., 2025). Yet, their black-box nature leaves a fundamental gap: While we can train models that classify signals with high accuracy, we often cannot tell why they made a specific decision. This opacity is especially concerning in high-stakes scenarios such as medical diagnosis or fault detection, where practitioners must understand the evidence behind a prediction before acting on it (Ro-

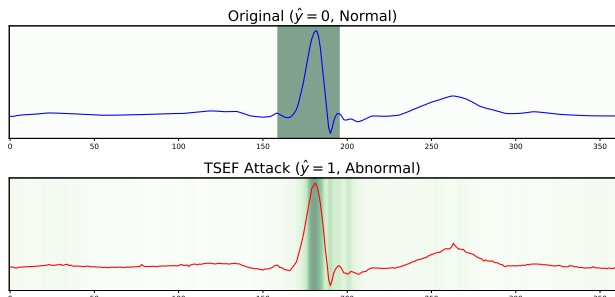

Original ($\hat{y} = 0$, Normal)

TSEF Attack ($\hat{y} = 1$, Abnormal)

*Figure 1.* Targeted manipulation of prediction and explanation on an ECG sample. **Top (Target Setup):** The input is correctly classified as Normal ($\hat{y} = 0$). The green band defines the reference explanation $\mathbf{A}'$ that the attacker aims to fabricate. **Bottom (TSEF Attack):** The proposed TSEF attack successfully flips the model's prediction to Abnormal ($\hat{y} = 1$). Crucially, the resulting explanation (generated by TIMEX++ (Liu et al., 2024c)) concentrates on the target region $\mathbf{A}'$ defined above.

jat et al., 2021; Theissler et al., 2022). To bridge this gap, interpretable time series deep learning systems (ITSDLS) pair a predictor with an explainer that highlights salient time–channel regions deemed responsible for the prediction (Queen et al., 2023; Liu et al., 2024c). In practice, such explanations are increasingly used for model debugging, expert verification, and even adversarial monitoring, providing an appealing layer of human oversight (Rojat et al., 2021; Šimić et al., 2021).

However, this workflow often relies on an implicit assumption: The explanation is faithful enough to be trusted by default. When a model raises an alarm, the accompanying explanation is frequently taken as a reliable rationale, rather than a hypothesis that must be stress-tested. Crucially, this default trust can be problematic: If the explanation is not faithful, it may provide a seemingly correct yet misleading justification, undermining human oversight and downstream decision-making. In vision and language, a growing body of work (Ghorbani et al., 2019; Zhang et al., 2020; Ivankay et al., 2022) has already shown that explanations can be misled by carefully crafted perturbations. Yet these vulnerabilities have not been systematically examined for time series models, whose inputs exhibit strong temporal dependencies (Ismail Fawaz et al., 2019), characteristic periodic or frequency-domain structures (Wu et al., 2023; Mu et al., 2025), and are routinely deployed in high-stakes settings (Lauritsen et al., 2020; Talukder et al., 2025; Je-Gal

[1]Emory University, USA [2]The Pennsylvania State University, USA [3]Michigan State University, USA [4]Griffith University, Australia. Correspondence to: Wei Jin <wei.jin@emory.edu>.

*Proceedings of the 43rd International Conference on Machine Learning*, Seoul, South Korea. PMLR 306, 2026. Copyright 2026 by the author(s).

et al., 2024; Feng et al., 2026; Liu et al., 2024b). For example, in ICU and ECG monitoring (Lauritsen et al., 2020; Talukder et al., 2025), clinicians and operators explicitly rely on explanations to validate alarms, debug unexpected behavior, and decide whether to act on a model's output.

At the same time, adversarial attacks have shown that time series classifiers can be fooled by imperceptible perturbations that change predictions (Mode & Hoque, 2020; Harford et al., 2020; Belkhouja & Doppa, 2022; Govindarajulu et al., 2023; Liu et al., 2023). The risk is further amplified when explanations are also subject to manipulation. If an attacker can simultaneously dictate what a model predicts and why it appears to predict it, interpretability will collapse from a safeguard to an illusion. Consider a clinical decision-support system: a malicious actor could subtly alter an ECG so that the classifier outputs a desired diagnosis, while the explainer highlights a convincing, but fabricated, physiological rationale. Figure 1 illustrates the severity of this threat: An attacker can jointly enforce a target label and a target attribution pattern. Such attacks can bypass human oversight and introduce downstream risk. This motivates the following question: *Can we trust the **joint** prediction–explanation output of an ITSDLS under adversarial perturbations?*

We answer this question by introducing TSEF (Time Series Explanation Fooler), an attack framework that crafts an adversarial example to (i) force a frozen classifier to a target class and (ii) steer its coupled explainer toward a reference explanation. This joint control is fundamentally different from attacking classifiers alone, and is not addressed by existing attacks on time series classification (Ding et al., 2023; Gu et al., 2025) or joint classification and explainer attacks in vision/NLP (Ghorbani et al., 2019; Zhang et al., 2020; Ivankay et al., 2022), for two modality-specific reasons. **First**, pattern-level control is essential: Time series models respond to temporal structures (e.g., trend, and periodic components), so naive per-timestep noise may not be strong enough to manipulate the model explanations or predictions. **Second**, time series exhibit a high-dimensional paradox: While the high-dimensional input offers a large norm-bounded attack surface, unconstrained optimization tends to produce dense, temporally incoherent updates that are difficult to steer toward a reference explanation, as revealed by our theoretical analysis in Section 4.1.

To resolve these challenges, TSEF restricts perturbations to a structured subspace governed by two components. A temporal vulnerability mask localizes *when* to edit through adaptive support reduction (sparse, connected segments), while a frequency filter determines *how* to edit by selecting *which* spectral content to modify, enabling controllable and contiguous explanation manipulation under the same budget. Consequently, our work exposes a critical prediction-interpretation gap in time series decision pipelines: An

attacker can simultaneously mislead both the model's prediction and its explanation, forcing a chosen label while making the explainer highlight a plausible rationale for the original class. Our contributions can be summarized as:

- **Systematic vulnerability study.** We provide the first assessment of adversarial robustness in time series explainers, showing they can be steered to match prescribed rationales even under target misclassification.
- **High-dimensional paradox for unstructured attacks.** We show that pointwise-bounded attacks disperse attribution mass as dimension grows, yielding diffuse explanations that fail to match sparse, contiguous target rationales.
- **TSEF: dual-target, structured attack.** We formulate a joint prediction–explanation objective and instantiate TSEF, decoupling *when* to perturb (sparse, connected temporal mask) from *how* to perturb (frequency-domain filter) under a perturbation budget.
- **Extensive validation and testbed.** We validate the threat across multiple datasets and interpreters, and release a modular testbed that standardizes coupled classifier–explainer evaluation under shared attack interfaces.

**Conflict of Interest Disclosure** The authors declare no financial conflicts of interest.

## 2. Related Work

**Time Series Explainers.** While Explainable AI (XAI) has been extensively studied for image and text (Danilevsky et al., 2020; Samek et al., 2021), explaining time series models presents unique challenges due to complex temporal dependencies (Rojat et al., 2021; Theissler et al., 2022). Early works adapted generic attribution methods (e.g., Integrated Gradients) to the temporal domain (Sundararajan et al., 2017; Lundberg & Lee, 2017; Ismail et al., 2020). Recent approaches develop time series specific explainers, such as perturbation-based methods (Crabbé & Van Der Schaar, 2021; Leung et al., 2023; Chuang et al., 2023; Huang et al., 2026) and mask-based methods like TIMEX and TIMEX++ (Queen et al., 2023; Liu et al., 2024c), which produce in-distribution and coherent explanations. **Our focus is orthogonal:** we study the *security* of these explainers under *adaptive adversaries*, showing that an attacker can enforce a target label while steering the explainer toward a *reference* explanation.

**Time Series Attacks.** Time series adversarial literature primarily targets *prediction robustness* in classification/forecasting, adapting image-style attacks and proposing structure-aware variants (e.g., region-focused or shapelet/high-frequency constrained perturbations) to improve stealth and effectiveness (Goodfellow et al., 2014; Chakraborty et al., 2018; Goyal et al., 2023; Karim et al., 2020; Ding et al., 2023; Gu et al., 2025; Mode & Hoque,

2020; Liu et al., 2023). A smaller line evaluates explainer quality/robustness via *non-adaptive* perturbation diagnostics (e.g., injecting random noise into regions highlighted by an explainer and measuring accuracy drop) (Nguyen et al., 2024). **Key gap:** existing methods either (i) attack predictions without controlling what explanations display, or (ii) probe explanations with random, non-targeted noise. **In contrast,** we formulate a *dual-target* objective that *simultaneously* forces a designated prediction and a designated explanation, exposing a failure mode that prediction-only robustness tests and noise-based explainer evaluations miss.

**Explainer Robustness in Vision and NLP.** Prior work in vision and NLP shows that explanations can be manipulated under small perturbations, either by selectively fooling attributions or jointly attacking predictions and explanations (Ghorbani et al., 2019; Zhang et al., 2020; Ivankay et al., 2022). However, achieving this in time series is nontrivial: effective attacks require *pattern-level* control, yet the $T \times D$ surface often yields dense, temporally incoherent updates that are difficult to steer toward a *target* explanation (Section 4.1). **Our contribution is specific to time series:** we instantiate a time–frequency structured dual-target attack tailored to temporal patterns, enabling targeted label manipulation while matching a reference rationale for explanation-based auditing. Further details are in Appendix A.

## 3. Preliminaries and Problem Definition

**Time Series Classifier.** A multivariate time series is a sequence of observations represented by a tensor $\mathbf{X} \in \mathbb{R}^{T \times D}$, where $T$ is the length of the sequence and $D$ is the number of features. A time series classifier, $f : \mathbb{R}^{T \times D} \to \mathbb{R}^{|\mathcal{C}|}$, is a function that maps an input time series $\mathbf{X}$ to a vector of logits or probabilities over a set of $|\mathcal{C}|$ classes. The final prediction is given by $\hat{y} = \arg\max f(\mathbf{X})$. We assume the classifier $f$ is pre-trained and differentiable.

**Time Series Explanations.** An explanation method, or explainer, $\mathcal{H}^E$, is a function that computes the importance of each feature at each time step for a prediction. For an input $\mathbf{X}$, the explainer outputs a saliency map $\mathbf{A} \in \mathbb{R}^{T \times D}$, where the magnitude of each element $\mathbf{A}[t, d]$ corresponds to the importance of the feature $d$ at time $t$. This general definition encompasses a wide range of explainers (Sundararajan et al., 2017; Crabbé & Van Der Schaar, 2021; Huang et al., 2026; Leung et al., 2023; Chuang et al., 2023), including TIMEX (Queen et al., 2023) and TIMEX++ (Liu et al., 2024c).

**Threat Model and Adversarial Objectives.** We operate under a white-box threat model, where the adversary has full knowledge of the target classifier $f$ and the explainer $\mathcal{H}^E$. The adversary's goal is to craft a small, norm-bounded perturbation $\delta \in \mathbb{R}^{T \times D}$ to generate an adversarial example

$\tilde{\mathbf{X}} = \mathbf{X} + \delta$. The perturbation is constrained to be small in the $L_\infty$ norm, i.e., $\|\delta\|_\infty \leq \epsilon$, to ensure it is not easily detectable. In this work, we investigate the adversarial goal: $\min_{\tilde{\mathbf{X}}} \ d\big(\mathcal{H}^E(\tilde{\mathbf{X}}), \mathbf{A}'\big)$ s.t. $f(\tilde{\mathbf{X}}) = y'$, $\|\mathbf{X} - \tilde{\mathbf{X}}\|_\infty \leq \epsilon$, where $\mathbf{X}$ is the original time series, $\tilde{\mathbf{X}}$ the attacked series, $f$ the classifier, $\mathcal{H}^E$ the explainer, $y'$ the target class, and $\mathbf{A}'$ the reference explanation. The function $d(\cdot, \cdot)$ measures discrepancy between explanations (e.g., MSE, cosine distance, KL divergence after normalization).

## 4. Method: TSEF

In this section, we first introduce the high-dimensional paradox challenge in attacking time series classifiers and explainers, and then propose our solution TSEF.

### 4.1. High-Dimensional Paradox

Standard adversarial attacks are typically *explanation-agnostic*: they use the full high-dimensional input space ($d = T \times D$) to optimize a classification objective only. Under an $\ell_\infty$ budget, these attacks apply small point-wise changes to many coordinates, and the accumulated effect can efficiently flip the prediction (Goodfellow et al., 2014). While effective for misclassification, we will show that this strategy creates a structural conflict with our goal of controlling the explainer. We quantify this conflict through the amount of attribution assigned *outside* the target region. Let $\Omega$ denote the small contiguous region supporting the reference explanation, and let $\mathbf{A}'$ be a hard mask strictly supported on $\Omega$ (where $|\Omega| \ll d$). Let $\mathbf{A}(\mathbf{X}) \in \mathbb{R}^d$ be the attribution map produced by an explainer, and let $\Omega^c$ denote the set of coordinates outside $\Omega$.

**Theorem 4.1** (Dense $\ell_\infty$ steps increase attribution outside the target region). *Let $g_c := \nabla_{\mathbf{X}} \mathcal{L}_{\mathrm{cls}}(f(\mathbf{X}), y_{\mathrm{tgt}})$ denote the gradient of the target-label classification loss, and consider the one-step dense $\ell_\infty$ update $\delta := -\varepsilon \, \mathrm{sign}(g_c)$ and $\tilde{\mathbf{X}} = \mathbf{X} + \delta$. Suppose the technical conditions in Appendix B hold, yielding constants $\gamma > 0$, $L > 0$, and $\beta_0 \geq 0$. Then for any $\varepsilon \in \left[\frac{4\beta_0}{\gamma}, \frac{\gamma}{2L}\right]$,*

$$\mathbb{E}\Big[\|\mathbf{A}(\tilde{\mathbf{X}})\|_{1,\Omega^c}\Big] \ \geq \ c\,\varepsilon\,(d - |\Omega|), \qquad (1)$$

*for some constant $c > 0$ independent of d. Consequently,*

$$\mathbb{E}\Big[\|\mathbf{A}(\tilde{\mathbf{X}}) - \mathbf{A}'\|_1\Big] \ \geq \ c\,\varepsilon\,(d - |\Omega|). \qquad (2)$$

We provide the proof of Theorem 4.1 in Appendix B. In this theorem, Eq. (1) states that after a single dense $\ell_\infty$ step, the total absolute attribution outside the target region $\Omega$ is lower bounded and scales with the number of coordinates outside $\Omega$; Eq. (2) then implies a corresponding lower bound on the overall mismatch to the reference explanation, which also grows with $(d - |\Omega|)$. This directly conflicts with the

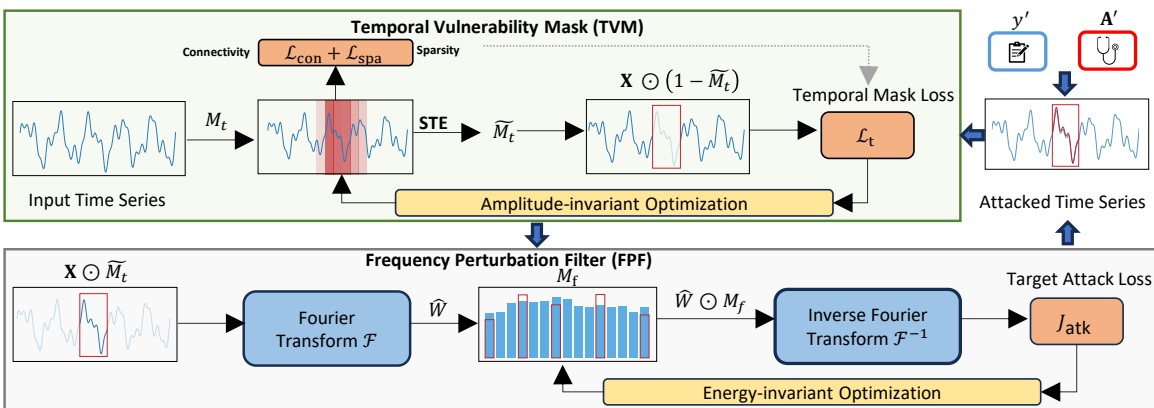

*Figure 2.* Overview of TSEF: TVM learns $\mathbf{M}_t$ to localize vulnerable temporal windows (visualized as a single window here for clarity), and FPF perturbs the masked signal in the frequency domain via $\mathbf{M}_f$.

form of explanations that look plausible for time series: they are typically localized, highlighting a short contiguous temporal structure (e.g., a brief event, trend, or periodic component) and showing little evidence elsewhere. A dense $\ell_\infty$ update modifies almost all coordinates, not only those in $\Omega$; even if each outside coordinate receives only a tiny amount of attribution, these contributions accumulate across many coordinates and yield a spread-out explanation.

**Motivation for TSEF.** A natural baseline is to optimize a joint objective $\mathcal{L}_{cls} + \lambda \mathcal{L}_{exp}$ under the same $\ell_\infty$ budget. However, the scaling in Eq. (2) highlights *high-dimensional* difficulty in time series: the classification term encourages changes spread across many coordinates (i.e, time steps and channels), while matching a localized reference rationale requires keeping the explanation concentrated within $\Omega$. To avoid the growth in Eq. (2), the explanation term would need to counteract the classification-driven updates across most coordinates outside $\Omega$, which becomes increasingly difficult as $d$ grows. Theorem 4.1 motivates attacks that explicitly control *where* changes are made. TSEF follows this principle by decoupling the attack into two components: (i) identify a small set of temporally connected segments that most strongly influence both the prediction and the explanation, and (ii) modify only these segments in a pattern-preserving manner. Frequency-domain editing is well-suited for (ii) because it induces coherent trend or periodicity changes in the time domain, avoiding scattered point-wise changes spread across the entire input.

### 4.2. TSEF Attack

To resolve the high-dimensional paradox, TSEF decouples the attack into *when* to attack (localize a vulnerable temporal pattern) and *how* to attack (apply a coherent spectral edit within that pattern). Concretely, TSEF learns (i) a temporal vulnerability mask ($\mathbf{M}_t$) that selects a time–channel region to manipulate, and (ii) a frequency perturbation filter ($\mathbf{M}_f$) that modifies the dynamics inside the selected region. We

learn $(\mathbf{M}_t, \mathbf{M}_f)$ by minimizing the following loss:

$$\min_{\mathbf{M}_f} \quad J_{atk}(\mathbf{M}_f; \mathbf{M}_t^*) := d\Big(\mathcal{H}^E\Big(\tilde{\mathbf{X}}(\mathbf{M}_t^*, \mathbf{M}_f)\Big), \mathbf{A}'\Big)$$

$$+ \lambda L_{cls}\Big(f\Big(\tilde{\mathbf{X}}(\mathbf{M}_t^*, \mathbf{M}_f)\Big), y'\Big)$$

$$\text{s.t.} \quad \mathbf{M}_t^* \in \arg\min_{\mathbf{M}_t} \quad d\big(\mathcal{H}^E(\mathbf{X} \odot (1 - \mathbf{M}_t)), \mathbf{A}'\big)$$

$$+ \lambda L_{cls}(f(\mathbf{X} \odot (1 - \mathbf{M}_t)), y').$$

We reconstruct the adversarial input by filtering the selected segment in the Fourier domain: $\tilde{\mathbf{X}}(\mathbf{M}_t, \mathbf{M}_f) = \mathcal{F}^{-1}(\mathcal{F}(\mathbf{M}_t \odot \mathbf{X}) \odot \mathbf{M}_f) + (1 - \mathbf{M}_t) \odot \mathbf{X}$. The inner loop localizes vulnerable patterns, while the outer loop refines the spectral perturbation under the $\ell_\infty$ budget. The pseudo algorithm is shown in Algorithm 1 in Appendix C.

#### 4.2.1. TEMPORAL VULNERABILITY MASK

A temporal pattern is *vulnerable* if suppressing it most effectively advances the *joint* objective (target prediction and reference explanation). Accordingly, the Temporal Vulnerability Mask $\mathbf{M}_t \in [0, 1]^{T \times D}$ parameterizes learnable *suppression probabilities* over time–channel locations to localize such patterns. We form the masked input $\mathbf{X}' = \mathbf{X} \odot (1 - \mathbf{M}_t)$, so that higher $\mathbf{M}_t[t, d]$ indicates stronger suppression at location $(t, d)$. We optimize $\mathbf{M}_t$ in the inner loop to localize a compact region whose removal most improves the joint target objective.

**Structured Mask: Sparsity and Connectivity.** Without regularization, the inner optimization may yield dense masks (trivially suppressing the entire series). We therefore regularize $\mathbf{M}_t$ to encourage sparse and contiguous patterns. For sparsity, we limit how much information the (stochastic) mask carries about the input. Let $M \in \{0, 1\}^{T \times D}$ be a binary mask random variable parameterized by $\mathbf{M}_t$ (e.g., independent Bernoulli entries), i.e., $P(M \mid \mathbf{X})$ is induced by the probabilities $\mathbf{M}_t(\mathbf{X})$. We minimize the mutual information $I(\mathbf{X}; M)$. Using a standard variational upper bound

with a prior $Q(M)$ independent of $\mathbf{X}$, we have

$$I(\mathbf{X}; M) \leq \mathbb{E}_{\mathbf{X}}[\text{KL}(P(M|\mathbf{X}) \,\|\, Q(M))], \quad (3)$$

with the derivation in Appendix D.1. In our parametric setting, $P(M|\mathbf{X})$ is defined by the probabilities $\mathbf{M}_t$. We instantiate the bound as an element-wise KL divergence against a sparse Bernoulli prior with rate $r$ (e.g., $r = 0.3$):

$$\mathcal{L}_{\text{spa}}(\mathbf{M}_t) = \frac{1}{TD} \sum_{t=1}^{T} \sum_{d=1}^{D} \text{KL}(\text{Bern}(\mathbf{M}_t[t, d]) \,\|\, \text{Bern}(r)).$$

To promote temporally contiguous patterns rather than scattered points, we further add a connectivity regularizer:

$$\mathcal{L}_{\text{con}}(\mathbf{M}_t) = \frac{1}{TD} \sum_{d=1}^{D} \sum_{t=1}^{T-1} (\mathbf{M}_t[t+1, d] - \mathbf{M}_t[t, d])^2.$$

**Full Inner Objective.** For each input $\mathbf{X}$, we optimize $\mathbf{M}_t$ by minimizing the joint adversarial and regularization loss:

$$\begin{aligned} \mathcal{L}_t(\mathbf{M}_t) = {} & \lambda_{\text{exp}}\, d(\mathcal{H}^E(\mathbf{X}'), \mathbf{A}') + \lambda_{\text{cls}}\, L_{\text{cls}}(f(\mathbf{X}'), y') \\ & + \lambda_{\text{spa}}\, \mathcal{L}_{\text{spa}}(\mathbf{M}_t) + \lambda_{\text{con}}\, \mathcal{L}_{\text{con}}(\mathbf{M}_t), \end{aligned}$$
$$(4)$$

where $\mathbf{X}' = \mathbf{X} \odot (1 - \mathbf{M}_t)$ is the masked input. The inner problem outputs the optimized mask $\mathbf{M}_t^*$, which localizes vulnerable temporal patterns for subsequent frequency-domain refinement.

**Amplitude-invariant Optimization.** A practical issue in time series is that amplitudes can vary substantially across channels and time steps; naive gradient updates may over-emphasize large-amplitude coordinates rather than truly sensitive ones. We therefore update the probability mask $\mathbf{M}_t$ using a projected sign step, $\mathbf{M}_t \leftarrow \Pi_{[0,1]}(\mathbf{M}_t - \eta_t \,\text{sign}(\nabla_{\mathbf{M}_t} \mathcal{L}_t))$, where $\Pi_{[0,1]}$ clips values to $[0, 1]$. As shown in Proposition D.1 (Appendix D.2), this update is amplitude-invariant with respect to the raw signal: whether $\mathbf{M}_t[t, d]$ increases or decreases depends on the directional alignment between $\mathbf{X}[t, d]$ and the loss gradient, rather than on $|\mathbf{X}[t, d]|$. This ensures the mask converges to structurally vulnerable regions rather than merely high-amplitude peaks.

**Implementation.** We implement discrete-like pattern selection using the Concrete (Gumbel–Sigmoid) relaxation with a straight-through estimator (STE) (Jang et al., 2016; Maddison et al., 2016). In the forward pass, we sample a hard binary mask $\tilde{\mathbf{M}}_t$ from $\mathbf{M}_t$ (Figure 2) and construct $\mathbf{X}' = \mathbf{X} \odot (1 - \tilde{\mathbf{M}}_t)$, while backpropagating gradients to the continuous parameters $\mathbf{M}_t$. For clarity, we write the masked input as $\mathbf{X}' = \mathbf{X} \odot (1 - \mathbf{M}_t)$ in the main text.

### 4.2.2. FREQUENCY PERTURBATION FILTER

Given the vulnerable temporal region localized by the optimized mask $\mathbf{M}_t^*$, the Frequency Perturbation Filter $\mathbf{M}_f$

specifies *how* to alter the underlying dynamics *within* that region. This is crucial for time series: while dense time-domain attacks can easily achieve misclassification, it often yields temporally incoherent artifacts and fails to synthesize a prescribed, contiguous attribution pattern. By operating in the frequency domain, $\mathbf{M}_f$ enables *pattern-consistent* modifications (e.g., trends and periodicities) that simultaneously steer prediction and explanation under the same $\ell_\infty$ budget.

**Forward Perturbation in Time–frequency Space.** Let $W := \mathbf{M}_t^* \odot \mathbf{X}$ denote the selected segment and $\widehat{W} := \mathcal{F}(W)$ its spectrum. We apply an element-wise multiplicative filter $\mathbf{M}_f$ in the frequency domain and reconstruct:

$$\widetilde{W} = \mathcal{F}^{-1}(\widehat{W} \odot \mathbf{M}_f), \quad \tilde{\mathbf{X}} = \widetilde{W} + (1 - \mathbf{M}_t^*) \odot \mathbf{X}.$$

This two-stage design decouples *when* to attack (via $\mathbf{M}_t^*$) from *how* to attack (via $\mathbf{M}_f$), and empirically yields stable, contiguous perturbations that match reference explanations.

**Parameterization and Feasibility.** We constrain the filter to $\mathbf{M}_f \in [0, 2]^{K \times D}$, where $\mathbf{M}_f = 1$ leaves a component unchanged, $\mathbf{M}_f < 1$ attenuates, and $\mathbf{M}_f > 1$ amplifies. To decouple the *spectral direction* from its *time-domain magnitude*, we parameterize

$$\mathbf{M}_f = \Pi_{[0,2]}(1 + \alpha_{\text{freq}} \tanh(\Theta_f)), \quad \alpha_{\text{freq}} \geq 0, \quad (5)$$

where $\Theta_f \in \mathbb{R}^{K \times D}$ is unconstrained and $\Pi_{[0,2]}$ clips entry-wise to $[0, 2]$. The scalar $\alpha_{\text{freq}}$ is chosen adaptively to satisfy the time-domain $\ell_\infty$ budget (see below). When working with complex spectra, we additionally enforce conjugate symmetry of $\mathbf{M}_f$ to guarantee a real-valued reconstruction.

**Adaptive Scaling.** Frequency-domain edits are *non-local* in time: small changes to a few spectral bins can produce large spikes after $\mathcal{F}^{-1}$, and the induced time-domain magnitude depends on the *instance-specific* window selected by $\mathbf{M}_t^*$. We therefore choose $\alpha_{\text{freq}}$ *adaptively* so that the perturbation induced on the current window respects the attack budget $\epsilon'$. Let $\Delta \mathbf{X}_{\text{base}} = \mathcal{F}^{-1}(\widehat{W} \odot \tanh(\Theta_f))$ denote the unit-direction time-domain change; applying Eq. (5) yields $\Delta \mathbf{X}(\alpha_{\text{freq}}) = \alpha_{\text{freq}} \Delta \mathbf{X}_{\text{base}}$ and thus $\|\Delta \mathbf{X}(\alpha_{\text{freq}})\|_\infty = \alpha_{\text{freq}} \|\Delta \mathbf{X}_{\text{base}}\|_\infty$. We set $\alpha_{\text{freq}} = \gamma \frac{\epsilon'}{\|\Delta \mathbf{X}_{\text{base}}\|_\infty + \tau}$, where $\gamma \in (0, 1)$ is a safety factor (we use $\gamma = 0.98$) and $\tau > 0$ ensures numerical stability. After each update of $\Theta_f$, we recompute $\alpha_{\text{freq}}$ and re-instantiate $\mathbf{M}_f$ via Eq. (5).

**Energy-invariant Optimization.** We update the frequency parameters $\Theta_f$ (which instantiate $\mathbf{M}_f$ via Eq. (5)) using a sign step: $\Theta_f \leftarrow \Theta_f - \eta_f \,\text{sign}(\nabla_{\Theta_f} J_{\text{atk}}(\tilde{\mathbf{X}}))$, where $J_{\text{atk}}$ is evaluated on the reconstructed $\tilde{\mathbf{X}}$. A key issue is that the gradient magnitude w.r.t. each frequency bin can scale with the spectral *magnitude* $|\widehat{W}[k, d]|$ (equivalently with $\sqrt{|\widehat{W}[k, d]|^2}$, where $|\widehat{W}[k, d]|^2$ is the bin energy), so standard gradient descent would be dominated by high-energy

bins. The sign operator removes this scale factor, making the update depend on the *directional correlation* between each bin and the attack objective rather than on its energy. This *energy-invariant* effect is formalized in Appendix D.3.

# 5. Experiment

## 5.1. Experimental Setup

**Datasets and Evaluation metrics.** We evaluate on six benchmarks: three synthetic datasets (LOWVAR, SEQCOMB-UV, SEQCOMB-MV) and three real-world datasets (ECG (Moody & Mark, 2001), PAM (Reiss & Stricker, 2012), Epilepsy (Andrzejak et al., 2001)). We use the train–test splits from prior work (Queen et al., 2023; Liu et al., 2024c) and attack correctly classified test samples. Dataset details are in Appendix E. For explanations, we report AUPRC, AUR, and AUP to quantify alignment between the produced importance maps and reference explanations. For classification, we report target-class F1 score and targeted Attack Success Rate (ASR), where ASR is the fraction of attacked samples whose predicted label equals the randomly chosen target label $y'$. Full metric definitions are provided in Appendix G.

**Threat models**: *(1) Time series interpreters.* We adopt TIMEX++ (Liu et al., 2024c), TIMEX (Queen et al., 2023), and Integrated Gradients (IG) (Sundararajan et al., 2017) as the time series explanation methods under attack. *(2) Time series classifier.* We use a transformer-based model (Vaswani et al., 2017) as the time series classifier for all datasets. **Baseline attacks and settings.** We compare against local/global Gaussian perturbations (Nguyen et al., 2024), Random Sign Attack, BlackTreeS (Ding et al., 2023), SFAttack (Gu et al., 2025), PGD (Madry et al., 2017), and ADV$^2$ (Zhang et al., 2020), a representative joint attack originally designed for image classifiers and explainers. We adapt baselines to the targeted setting by optimizing a targeted classification toward $y'$; To adapt ADV$^2$ to time series, we replace its interpretation loss with a time series explanation-alignment term toward $\mathbf{A}'$, so that all methods are evaluated under the same targeted objectives as TSEF. All attacks use an $\ell_\infty$ budget $\epsilon = 0.1$. To handle heterogeneous signal scales, we use an instance-wise bound $\epsilon' = \epsilon \cdot (\max(\mathbf{X}) - \min(\mathbf{X}))$ and enforce $\|\mathbf{X} - \tilde{\mathbf{X}}\|_\infty \leq \epsilon'$ via projection at each step (Alg. 1). Hyperparameters are tuned on validation sets; full settings are in Appendix F.

**Target objectives.** For each correctly classified test instance with label $y$, we sample $y' \neq y$ uniformly from the remaining classes and optimize a dual-target objective that (i) drives the predictor toward $y'$ and (ii) forces the explanation to match a reference explanation mask $\mathbf{A}'$. When temporal annotations are available, $\mathbf{A}'$ is the ground-truth mask; otherwise, we set $\mathbf{A}'$ to the top-$k\%$ time steps from

the *clean* explanation of the same interpreter.

**Additional analyses.** Additional experiments on surrogate transferability, alternative classifier backbones, the original-explanation-preserving setting, hyperparameter sensitivity, and additional ablations are reported in Appendix H.

## 5.2. Attack Effectiveness on Predictions/Explanations

**Threat model: dual-target attack for auditing.** We study a realistic failure mode of ITSDLS: an adversary forces the prediction to a target label $y'$ while keeping the explanation aligned with the original-class rationale $\mathbf{A}_{\mathrm{orig}}$. A successful attack satisfies $f(\tilde{\mathbf{X}}) = y'$ and preserves alignment to $\mathbf{A}_{\mathrm{orig}}$; we report ASR/F1 and explanation alignment (AUPRC, AUP, AUR). We compare against prediction-only attacks (PGD, BlackTreeS, SFAttack), a dual-objective baseline (ADV$^2$), and sanity-check perturbations (Random, Local/Global Gaussian). Table 1 summarizes results on four benchmarks and three interpreters. Our goal is not to crown the strongest prediction attack, but to stress-test a common auditing premise: *stable, plausible explanations imply a stable, trustworthy decision.*

**(1) Explanation-preserving attack is feasible without sacrificing targeted prediction.** Across all datasets, TSEF maintains strong targeting success *while enforcing explanation constraints*. For instance, on LOWVAR with TIMEX++, TSEF attains $0.837/0.848$ (F1/ASR), comparable to PGD ($0.833/0.846$). Similar trends hold on SEQCOMB-MV, and on ECG the attack keeps ASR near the strongest baselines (TIMEX++ ASR $0.951$ vs. $0.946$ for PGD). These results establish the *existence* of explanation-preserving attacks: explanation constraints do not inherently prevent targeted prediction manipulation.

**(2) Vulnerability emerges in explanation stability.** While several baselines appear effective in ASR, their success is superficial: explanations typically drift significantly from $\mathbf{A}_{\mathrm{orig}}$, acting as a warning signal for auditors. TSEF exposes the true robustness gap by closing this loop. For instance, on LOWVAR with TIMEX++, PGD reaches $0.846$ ASR but yields only $0.258$ AUPRC, whereas TSEF preserves both high ASR ($0.848$) and high alignment (AUPRC $0.845$; AUP $0.760$; AUR $0.800$). On SEQCOMB-MV, TSEF increases TIMEX++ AUPRC from $0.484$ (PGD) to $0.661$ at similar ASR. Even on ECG, where prediction targeting is already easy for many attacks, TSEF still raises the explanation metrics under TIMEX++. Crucially, these gains do not aim to be best overall on every scalar metric; they demonstrate a more concerning phenomenon: *the system can be driven to a wrong decision while the explanation remains consistent enough to pass an explanation-based audit.*

**(3) System-level view: A failure of coupled robustness.** Table 1 reveals a trade-off for existing attacks: (i) prediction-

*Table 1.* Attack performance on four benchmark datasets. ASR measures the attack success rate on the target predictions, while F1 denotes the F1 score for the target class. AUPRC, AUP, and AUR quantify attack effectiveness on the explanations.

| INTERPRETER | ATTACK | LowVar | | | | | SeqComb-UV | | | | |
|---|---|---|---|---|---|---|---|---|---|---|---|
| | | F1↑ | ASR↑ | AUPRC↑ | AUP↑ | AUR↑ | F1↑ | ASR↑ | AUPRC↑ | AUP↑ | AUR↑ |
| TimeX++ | PGD | 0.833±0.040 | 0.846±0.036 | 0.258±0.005 | 0.194±0.004 | 0.318±0.004 | 0.514±0.031 | 0.502±0.028 | 0.465±0.004 | 0.552±0.005 | 0.575±0.004 |
| | BlackTrees | 0.728±0.047 | 0.740±0.045 | 0.361±0.005 | 0.271±0.004 | 0.375±0.004 | 0.533±0.034 | 0.518±0.031 | 0.484±0.005 | 0.566±0.005 | 0.573±0.004 |
| | SFAttack | 0.768±0.040 | 0.781±0.037 | 0.336±0.005 | 0.242±0.004 | 0.380±0.004 | 0.474±0.041 | 0.460±0.039 | 0.488±0.005 | 0.567±0.005 | 0.574±0.004 |
| | ADV$^2$ | 0.777±0.039 | 0.795±0.033 | 0.617±0.005 | 0.522±0.005 | 0.609±0.004 | 0.483±0.038 | 0.468±0.037 | 0.533±0.005 | 0.571±0.005 | 0.600±0.004 |
| | Random | 0.014±0.001 | 0.014±0.001 | 0.786±0.004 | 0.643±0.004 | 0.730±0.003 | 0.027±0.002 | 0.027±0.002 | 0.471±0.004 | 0.549±0.005 | 0.585±0.004 |
| | Local G | 0.050±0.006 | 0.061±0.008 | 0.716±0.004 | 0.602±0.004 | 0.661±0.003 | 0.010±0.001 | 0.010±0.001 | 0.484±0.005 | 0.567±0.005 | 0.559±0.004 |
| | Global G | 0.051±0.004 | 0.062±0.006 | 0.718±0.005 | 0.604±0.004 | 0.664±0.003 | 0.010±0.002 | 0.010±0.002 | 0.484±0.005 | 0.567±0.005 | 0.559±0.004 |
| | **TSEF (Ours)** | **0.837±0.046** | **0.848±0.042** | **0.845±0.004** | **0.760±0.004** | **0.800±0.003** | **0.584±0.037** | **0.568±0.035** | **0.577±0.006** | **0.578±0.005** | **0.620±0.004** |
| TimeX | PGD | **0.833±0.039** | **0.848±0.034** | 0.213±0.004 | 0.128±0.003 | 0.443±0.004 | 0.503±0.030 | 0.488±0.028 | 0.422±0.004 | 0.633±0.005 | 0.491±0.005 |
| | BlackTrees | 0.732±0.046 | 0.747±0.043 | 0.291±0.005 | 0.180±0.003 | 0.514±0.004 | 0.528±0.036 | 0.512±0.036 | 0.431±0.004 | 0.640±0.006 | 0.487±0.005 |
| | SFAttack | 0.759±0.044 | 0.770±0.042 | 0.309±0.004 | 0.182±0.003 | 0.564±0.004 | 0.478±0.041 | 0.470±0.040 | 0.428±0.004 | 0.638±0.006 | 0.486±0.005 |
| | ADV$^2$ | 0.765±0.047 | 0.788±0.039 | 0.464±0.005 | 0.321±0.004 | 0.640±0.004 | 0.493±0.031 | 0.478±0.031 | 0.493±0.005 | 0.645±0.006 | 0.524±0.004 |
| | Random | 0.011±0.001 | 0.011±0.001 | 0.632±0.005 | 0.441±0.004 | **0.818±0.003** | 0.025±0.002 | 0.025±0.002 | 0.433±0.004 | 0.632±0.006 | 0.511±0.005 |
| | Local G | 0.046±0.009 | 0.058±0.013 | 0.634±0.005 | 0.492±0.004 | 0.730±0.003 | 0.013±0.002 | 0.013±0.002 | 0.427±0.004 | 0.634±0.006 | 0.463±0.005 |
| | Global G | 0.046±0.010 | 0.056±0.012 | 0.632±0.005 | 0.490±0.004 | 0.730±0.003 | 0.017±0.002 | 0.017±0.002 | 0.427±0.004 | 0.633±0.006 | 0.463±0.005 |
| | **TSEF (Ours)** | 0.818±0.055 | 0.835±0.048 | **0.751±0.005** | **0.597±0.004** | 0.811±0.003 | **0.544±0.036** | **0.525±0.034** | **0.543±0.005** | **0.649±0.006** | **0.552±0.004** |
| Integrated Gradients | PGD | 0.825±0.045 | 0.843±0.039 | 0.191±0.004 | 0.097±0.003 | 0.569±0.003 | 0.502±0.037 | 0.487±0.036 | 0.304±0.003 | 0.435±0.005 | 0.611±0.004 |
| | BlackTrees | 0.725±0.048 | 0.737±0.045 | 0.258±0.004 | 0.146±0.003 | 0.561±0.003 | 0.508±0.038 | 0.493±0.037 | 0.330±0.004 | 0.467±0.005 | 0.589±0.004 |
| | SFAttack | 0.759±0.039 | 0.774±0.036 | 0.239±0.005 | 0.123±0.003 | 0.566±0.003 | 0.458±0.038 | 0.446±0.035 | 0.317±0.003 | 0.440±0.005 | 0.611±0.004 |
| | ADV$^2$ | 0.821±0.044 | 0.838±0.039 | 0.417±0.007 | 0.250±0.004 | 0.642±0.003 | 0.490±0.035 | 0.476±0.033 | 0.374±0.004 | 0.438±0.005 | 0.625±0.004 |
| | Random | 0.014±0.003 | 0.014±0.003 | **0.695±0.005** | **0.468±0.004** | 0.645±0.002 | 0.023±0.003 | 0.024±0.003 | 0.336±0.003 | **0.539±0.005** | 0.516±0.005 |
| | Local G | 0.054±0.004 | 0.067±0.007 | 0.367±0.006 | 0.222±0.004 | 0.581±0.002 | 0.032±0.003 | 0.032±0.003 | 0.333±0.004 | 0.482±0.005 | 0.574±0.005 |
| | Global G | 0.060±0.005 | 0.073±0.007 | 0.365±0.006 | 0.221±0.004 | 0.582±0.002 | 0.039±0.003 | 0.039±0.003 | 0.333±0.004 | 0.482±0.005 | 0.574±0.005 |
| | **TSEF (Ours)** | **0.843±0.030** | **0.852±0.027** | 0.545±0.007 | 0.341±0.005 | **0.669±0.003** | **0.564±0.033** | **0.546±0.031** | **0.437±0.004** | 0.444±0.005 | **0.625±0.004** |

| INTERPRETER | ATTACK | SeqComb-MV | | | | | ECG | | | | |
|---|---|---|---|---|---|---|---|---|---|---|---|
| | | F1↑ | ASR↑ | AUPRC↑ | AUP↑ | AUR↑ | F1↑ | ASR↑ | AUPRC↑ | AUP↑ | AUR↑ |
| TimeX++ | PGD | 0.752±0.021 | 0.745±0.023 | 0.484±0.004 | 0.619±0.005 | 0.527±0.004 | 0.902±0.018 | 0.946±0.011 | 0.639±0.001 | 0.715±0.001 | 0.431±0.001 |
| | BlackTrees | 0.771±0.021 | 0.765±0.022 | 0.524±0.004 | 0.639±0.005 | 0.532±0.004 | 0.882±0.023 | 0.934±0.015 | 0.650±0.001 | 0.737±0.001 | 0.427±0.001 |
| | SFAttack | 0.709±0.030 | 0.698±0.032 | 0.462±0.004 | 0.606±0.005 | 0.522±0.004 | 0.892±0.019 | 0.941±0.011 | 0.647±0.001 | 0.721±0.001 | 0.432±0.001 |
| | ADV$^2$ | 0.742±0.019 | 0.735±0.022 | 0.630±0.006 | **0.645±0.006** | 0.607±0.004 | 0.887±0.013 | 0.935±0.013 | 0.705±0.001 | 0.773±0.001 | 0.448±0.001 |
| | Random | 0.065±0.015 | 0.090±0.023 | 0.452±0.004 | 0.571±0.004 | 0.546±0.004 | 0.090±0.009 | 0.093±0.010 | 0.636±0.001 | 0.724±0.001 | 0.427±0.001 |
| | Local G | 0.015±0.002 | 0.015±0.002 | 0.428±0.004 | 0.641±0.005 | 0.479±0.005 | 0.053±0.004 | 0.054±0.004 | 0.652±0.001 | 0.742±0.001 | 0.419±0.001 |
| | Global G | 0.015±0.002 | 0.015±0.002 | 0.427±0.004 | 0.640±0.005 | 0.479±0.005 | 0.053±0.006 | 0.054±0.005 | 0.657±0.001 | 0.749±0.001 | 0.422±0.001 |
| | **TSEF (Ours)** | **0.795±0.012** | **0.789±0.015** | **0.661±0.006** | 0.641±0.005 | **0.631±0.003** | **0.911±0.017** | **0.951±0.010** | **0.713±0.001** | **0.776±0.001** | **0.450±0.001** |
| TimeX | PGD | 0.739±0.018 | 0.731±0.020 | 0.154±0.002 | 0.263±0.004 | 0.527±0.004 | **0.902±0.018** | **0.946±0.011** | 0.320±0.001 | 0.345±0.001 | **0.475±0.001** |
| | BlackTrees | 0.771±0.023 | 0.764±0.025 | 0.164±0.002 | 0.280±0.005 | 0.527±0.004 | 0.882±0.023 | 0.934±0.015 | 0.353±0.001 | 0.376±0.001 | 0.468±0.001 |
| | SFAttack | 0.703±0.033 | 0.692±0.036 | 0.149±0.002 | 0.262±0.004 | 0.524±0.004 | 0.892±0.019 | 0.941±0.011 | 0.327±0.001 | 0.361±0.001 | 0.463±0.001 |
| | ADV$^2$ | 0.744±0.019 | 0.736±0.021 | 0.221±0.003 | 0.292±0.005 | 0.567±0.004 | 0.887±0.015 | 0.937±0.010 | 0.396±0.001 | 0.424±0.001 | 0.456±0.001 |
| | Random | 0.068±0.013 | 0.089±0.020 | 0.144±0.002 | 0.231±0.004 | 0.533±0.004 | 0.090±0.009 | 0.092±0.010 | 0.317±0.001 | 0.345±0.001 | 0.471±0.001 |
| | Local G | 0.009±0.004 | 0.009±0.004 | 0.138±0.002 | 0.271±0.004 | 0.513±0.004 | 0.062±0.007 | 0.063±0.007 | 0.347±0.001 | 0.380±0.001 | 0.451±0.001 |
| | Global G | 0.007±0.003 | 0.008±0.003 | 0.137±0.002 | 0.270±0.004 | 0.513±0.004 | 0.062±0.008 | 0.062±0.008 | 0.333±0.001 | 0.357±0.001 | 0.452±0.001 |
| | **TSEF (Ours)** | **0.803±0.013** | **0.796±0.015** | **0.235±0.003** | **0.295±0.005** | **0.576±0.004** | 0.899±0.017 | 0.943±0.011 | **0.419±0.001** | **0.446±0.001** | 0.468±0.001 |
| Integrated Gradients | PGD | 0.734±0.021 | 0.726±0.024 | 0.183±0.002 | 0.349±0.004 | 0.607±0.004 | 0.902±0.018 | 0.946±0.011 | 0.199±0.000 | 0.247±0.001 | 0.697±0.001 |
| | BlackTrees | 0.766±0.022 | 0.759±0.024 | 0.192±0.002 | 0.368±0.005 | 0.592±0.004 | 0.871±0.023 | 0.925±0.015 | 0.329±0.001 | **0.347±0.001** | 0.621±0.001 |
| | SFAttack | 0.708±0.028 | 0.698±0.030 | 0.162±0.002 | 0.319±0.004 | 0.606±0.004 | 0.892±0.019 | 0.941±0.011 | 0.199±0.000 | 0.241±0.001 | 0.701±0.001 |
| | ADV$^2$ | 0.741±0.018 | 0.733±0.020 | 0.281±0.003 | 0.363±0.004 | **0.641±0.004** | 0.902±0.020 | 0.946±0.012 | 0.276±0.001 | 0.288±0.001 | **0.715±0.001** |
| | Random | 0.069±0.012 | 0.093±0.020 | 0.174±0.002 | 0.375±0.004 | 0.541±0.005 | 0.090±0.009 | 0.093±0.010 | 0.197±0.000 | 0.264±0.001 | 0.559±0.001 |
| | Local G | 0.018±0.004 | 0.018±0.004 | 0.170±0.002 | 0.317±0.004 | 0.624±0.004 | 0.103±0.016 | 0.104±0.016 | 0.227±0.001 | 0.297±0.001 | 0.553±0.001 |
| | Global G | 0.019±0.003 | 0.019±0.004 | 0.169±0.002 | 0.316±0.004 | 0.621±0.004 | 0.119±0.015 | 0.120±0.014 | 0.218±0.000 | 0.275±0.001 | 0.570±0.001 |
| | **TSEF (Ours)** | **0.789±0.012** | **0.781±0.014** | **0.334±0.004** | **0.376±0.004** | 0.638±0.004 | **0.909±0.023** | **0.949±0.014** | **0.354±0.001** | 0.321±0.001 | 0.644±0.001 |

*Table 2.* Ablation study on the SeqComb-MV dataset.

| INTERPRETER | ATTACK | F1↑ | ASR↑ | AUPRC↑ | AUP↑ | AUR↑ |
|---|---|---|---|---|---|---|
| TimeX++ | TSEF (w/o $M_t$) | 0.508±0.020 | 0.491±0.021 | 0.551±0.005 | 0.621±0.005 | 0.578±0.004 |
| | TSEF (w/o $M_f$) | 0.686±0.016 | 0.678±0.018 | 0.632±0.006 | 0.639±0.005 | 0.614±0.003 |
| | **TSEF (Full)** | **0.795±0.012** | **0.789±0.015** | **0.661±0.006** | **0.641±0.005** | **0.631±0.003** |
| Integrated Gradients | TSEF (w/o $M_t$) | 0.559±0.027 | 0.544±0.027 | 0.247±0.003 | 0.353±0.004 | 0.627±0.004 |
| | TSEF (w/o $M_f$) | 0.672±0.030 | 0.660±0.033 | 0.253±0.003 | 0.368±0.004 | 0.626±0.004 |
| | **TSEF (Full)** | **0.789±0.012** | **0.781±0.014** | **0.334±0.004** | **0.376±0.004** | **0.638±0.004** |

only methods achieve high ASR but low explanation alignment, (ii) random/noise perturbations can yield non-trivial explanation scores yet rarely reach $y'$, and (iii) joint optimization (ADV$^2$) partially reduces explanation drift yet remains behind TSEF on alignment at comparable targeting. The key takeaway is that robustness in ITSDLS must be evaluated in the *coupled predictor–explainer space*. An auditor relying on explanation stability as a guardrail against manipulation can be systematically misled.

### 5.3. Attack Effectiveness on Different Interpreters

We further investigate whether this cover-up vulnerability is specific to an explanation family. By varying the explanation mechanism (TimeX/TimeX++ vs. Integrated Gradients), we find the vulnerability is systemic.

**Mask-based explainers (TimeX/TimeX++).** These methods use explicit regularization (smoothness, information bottleneck) to stabilize masks. However, our results on ECG show that TSEF improves AUPRC from $0.320$ (PGD) to $0.419$ (TimeX) while maintaining $94\%$ ASR. Under TimeX++, TSEF simultaneously achieves high ASR ($95\%$) and strong alignment (AUPRC $0.713$), indicating that regularized mask explanations can still be *steered* without revealing obvious drift. **Gradient-based explainers (Integrated Gradients).** IG is generally harder to control due to path integration and baseline sensitivity (Sundararajan et al., 2017; Sturmfels et al., 2020), and we indeed observe lower absolute alignment scores than mask-based explainers. However, TSEF still demonstrates the capacity to decouple prediction from explanation: on SeqComb-MV (IG), it improves AUPRC/AUP to $0.334/0.376$ compared to $0.183/0.349$ (PGD), while keeping strong ASR. This confirms that the deceptive alignment risk generalizes beyond mask-based ex-

*Table 3.* Attack performance on the PAM and Epilepsy datasets. Best results are highlighted in bold.

| INTERPRETER | ATTACK | PAM | | | | | EPILEPSY | | | | |
|---|---|---|---|---|---|---|---|---|---|---|---|
| | | F1 ↑ | ASR ↑ | AUPRC ↑ | AUP ↑ | AUR ↑ | F1 ↑ | ASR ↑ | AUPRC ↑ | AUP ↑ | AUR ↑ |
| TIMEX++ | ADV$^2$ | $0.594 \pm 0.016$ | $0.585 \pm 0.017$ | $0.820 \pm 0.004$ | $0.702 \pm 0.003$ | $0.474 \pm 0.003$ | $0.159 \pm 0.011$ | $0.190 \pm 0.016$ | $0.837 \pm 0.005$ | $0.381 \pm 0.004$ | $0.894 \pm 0.002$ |
| | PGD | $0.604 \pm 0.019$ | $0.597 \pm 0.019$ | $0.696 \pm 0.006$ | $0.641 \pm 0.004$ | $0.446 \pm 0.004$ | $0.160 \pm 0.011$ | $0.191 \pm 0.016$ | $0.658 \pm 0.006$ | $0.345 \pm 0.004$ | $0.864 \pm 0.002$ |
| | **TSEF (OURS)** | $\mathbf{0.624 \pm 0.015}$ | $\mathbf{0.614 \pm 0.015}$ | $\mathbf{0.876 \pm 0.003}$ | $\mathbf{0.729 \pm 0.003}$ | $\mathbf{0.490 \pm 0.003}$ | $\mathbf{0.165 \pm 0.012}$ | $\mathbf{0.199 \pm 0.018}$ | $\mathbf{0.912 \pm 0.005}$ | $\mathbf{0.422 \pm 0.005}$ | $\mathbf{0.907 \pm 0.002}$ |
| TIMEX | ADV$^2$ | $0.579 \pm 0.018$ | $0.575 \pm 0.020$ | $0.743 \pm 0.005$ | $0.760 \pm 0.003$ | $0.364 \pm 0.003$ | $0.146 \pm 0.010$ | $0.171 \pm 0.014$ | $0.922 \pm 0.003$ | $0.583 \pm 0.003$ | $0.841 \pm 0.002$ |
| | PGD | $0.600 \pm 0.018$ | $0.589 \pm 0.018$ | $0.563 \pm 0.006$ | $0.674 \pm 0.004$ | $0.318 \pm 0.003$ | $\mathbf{0.160 \pm 0.011}$ | $\mathbf{0.191 \pm 0.016}$ | $0.876 \pm 0.003$ | $0.550 \pm 0.003$ | $0.806 \pm 0.002$ |
| | **TSEF (OURS)** | $\mathbf{0.611 \pm 0.018}$ | $\mathbf{0.599 \pm 0.018}$ | $\mathbf{0.836 \pm 0.003}$ | $\mathbf{0.798 \pm 0.002}$ | $\mathbf{0.395 \pm 0.002}$ | $0.158 \pm 0.016$ | $0.190 \pm 0.023$ | $\mathbf{0.981 \pm 0.002}$ | $\mathbf{0.652 \pm 0.003}$ | $\mathbf{0.859 \pm 0.002}$ |
| INTEGRATED GRADIENTS | ADV$^2$ | $0.600 \pm 0.018$ | $0.591 \pm 0.017$ | $0.681 \pm 0.005$ | $0.583 \pm 0.004$ | $\mathbf{0.464 \pm 0.004}$ | $\mathbf{0.160 \pm 0.011}$ | $\mathbf{0.192 \pm 0.016}$ | $0.629 \pm 0.007$ | $0.491 \pm 0.007$ | $0.565 \pm 0.005$ |
| | PGD | $0.592 \pm 0.015$ | $0.585 \pm 0.016$ | $0.646 \pm 0.005$ | $0.576 \pm 0.004$ | $0.461 \pm 0.004$ | $0.160 \pm 0.011$ | $0.191 \pm 0.016$ | $0.623 \pm 0.007$ | $0.489 \pm 0.007$ | $0.563 \pm 0.005$ |
| | **TSEF (OURS)** | $\mathbf{0.603 \pm 0.021}$ | $\mathbf{0.594 \pm 0.023}$ | $\mathbf{0.713 \pm 0.004}$ | $\mathbf{0.604 \pm 0.004}$ | $0.454 \pm 0.004$ | $0.160 \pm 0.016$ | $0.192 \pm 0.022$ | $\mathbf{0.663 \pm 0.007}$ | $\mathbf{0.515 \pm 0.008}$ | $\mathbf{0.569 \pm 0.005}$ |

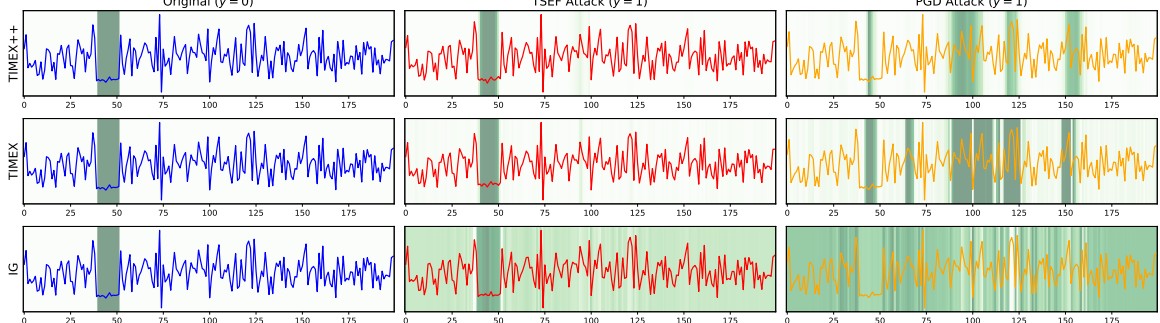

*Figure 3.* **Targeted manipulation of prediction and explanation on a LOWVAR sample.** Rows correspond to TIMEX++, TIMEX, and Integrated Gradients (IG). **Left:** Clean input with prediction $\hat{y} = 0$; the dark green window marks the reference explanation region. **Middle:** TSEF flips the prediction to the target class ($\hat{y} = 1$) while keeping the explanation concentrated on the same target region. **Right:** Standard PGD also attains $\hat{y} = 1$ but produces diffuse, fragmented attributions spread across time.

planation to gradient-based attribution. Across four benchmarks and three interpreters, TSEF reveals a concrete failure mode: *predictions can be compromised while explanations remain stable and original-looking.* This highlights the need for **future explainer development to explicitly evaluate adversarial robustness**.

### 5.4. Vulnerability in Real-World Deployments

Our benchmark results reveal a system-level failure mode. We now test whether this vulnerability persists in real deployments where ground-truth rationales are unavailable and the clean explanation serves as the auditing reference. We define the reference rationale $\mathbf{A}'$ as the interpreter's top-10% salient features on the clean input, and optimize the dual-target objective to preserve this reference while forcing $y'$. Table 3 shows that TSEF achieves a stronger prediction–explanation decoupling than PGD and ADV$^2$ on both datasets, maintaining comparable targeting while improving explanation alignment across interpreters. For example, under TIMEX++ on PAM, TSEF attains ASR $0.614$ while increasing AUPRC to $0.876$ (vs. $0.585/0.820$ for ADV$^2$ and $0.597/0.696$ for PGD). On Epilepsy, it similarly strengthens preservation (e.g., TIMEX AUPRC $0.981$ vs. $0.922/0.876$, and IG AUPRC $0.663$ vs. $0.629/0.623$). **Implication for explanation-based auditing:** In practice, explanation drift is treated as an alarm: predictions are trusted when explanations remain consistent with a pre-attack reference. Our results challenge this premise: targeted misclassification can be induced while the explanation stays highly consistent

with the reference, allowing an attacker to evade monitoring.

### 5.5. Ablation Study

We ablate TSEF's temporal vulnerability mask $\mathbf{M}_t$ and frequency perturbation filter $\mathbf{M}_f$ via **w/o $\mathbf{M}_t$** (global spectral update) and **w/o $\mathbf{M}_f$** (localized PGD update). Table 2 shows that the full method achieves the best *joint* trade-off on SEQCOMB-MV across interpreters, combining strong targeting (F1/ASR) with improved explanation alignment (AUPRC/AUP/AUR). Removing $\mathbf{M}_t$ substantially hurts targeting (e.g., SEQCOMB-MV+TIMEX++: ASR $0.789 \rightarrow 0.491$), highlighting the need for temporal localization under $\ell_\infty$. Removing $\mathbf{M}_f$ also degrades performance (e.g., SEQCOMB-MV+IG: AUPRC $0.334 \rightarrow 0.253$). Overall, both $\mathbf{M}_t$ and $\mathbf{M}_f$ are necessary to satisfy the dual objectives. Similar observations are shown in Appendix H.5.

### 5.6. Visualization

Figure 3 corroborates our quantitative findings. We visualize the setting, where the attacker sets the *reference explanation* to be the *clean rationale* (green window). The middle column shows that TSEF flips the prediction to the target label ($\hat{y} = 1$) *while keeping the explanation aligned with the reference window*, yielding a sharp and concentrated heatmap. This visual consistency explains the high AUPRC in Table 1, as the attack recovers the reference rationale with high precision. In contrast, PGD attains the same target prediction but produces diffuse attributions across time.

Such dispersion may still overlap the window, yet it severely degrades precision and introduces conspicuous artifacts that undermine the cover-up under explanation-based auditing. Overall, TSEF achieves *dual-target control* over *what* the model predicts and *why* it appears to predict it.

## 6. Conclusion

This work exposes a systemic vulnerability in interpretable time series systems: the decoupling between prediction and explanation enables attackers to alter the former while camouflaging the latter. Through TSEF, we show that explanation stability is a misleading proxy for robustness; thus, auditing protocols must move beyond simple consistency checks. Future research should develop coupling-aware defenses and audits that explicitly constrain predictor–explainer behavior, and extend evaluations to realistic, constrained threat models to restore trust in high-stakes settings.

**Limitations and scope.** TSEF is primarily a white-box stress test of coupled predictor–explainer robustness and requires gradients of both the classifier and the explainer. Therefore, the current formulation directly applies to differentiable explainers, while perturbation-, sampling-, or surrogate-based explainers require different optimization paradigms, such as surrogate-transfer or query-based attacks. Our transfer experiments in Appendix H provide preliminary unseen-target evidence, but fully black-box joint attacks and defenses remain important future directions.

## Acknowledgement

This research was supported by the U.S. National Science Foundation under Award Nos. 2504088, 2437345, 2302968, and 2124104, the National Institute of Health under Award Nos. R01ES033241 and R01LM013712.

## Impact Statement

This paper presents work whose goal is to advance the field of Machine Learning. There are many potential societal consequences of our work, none which we feel must be specifically highlighted here.

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

# A. Related Work Details

This appendix complements the concise Related Work in the main paper by providing additional context and representative references. We organize the discussion around three threads: (i) time series explainers, (ii) adversarial attacks on time series models and robustness diagnostics for explainers, and (iii) explainer robustness in vision and NLP. Throughout, we highlight how our threat model and objective (dual-target, cover-up manipulation) differs from prior settings.

**Time series analysis and time series explainers.**    Time series analysis plays a foundational role across a wide range of domains such as finance (Sezer et al., 2020; Emami Gohari et al., 2024), traffic systems (Wu et al., 2021; Zhou et al., 2022; Liu et al., 2024a; 2025b), healthcare (Talukder et al., 2025; Wang et al., 2026a;b), epidemiology (Liu et al., 2024b; Wan et al., 2025), and audio processing (Baevski et al., 2020; Wang et al., 2023). Recent work also broadens time series analysis toward efficient long-range temporal modeling and spatio-temporal reasoning (Ni et al., 2025; 2026a;b). The increasing deployment of deep models in high-stakes applications has made human-understandable explanations a key requirement for trustworthy decision making. While the broader XAI literature is dominated by vision and language settings (Danilevsky et al., 2020; Madsen et al., 2022; Bodria et al., 2023; Linardatos et al., 2020; Samek et al., 2021; Buhrmester et al., 2021), time series interpretability has received comparatively less attention despite its practical importance (Rojat et al., 2021; Theissler et al., 2022). Early efforts often adapt generic attribution tools such as Integrated Gradients and SHAP to temporal inputs (Sundararajan et al., 2017; Lundberg & Lee, 2017). However, time series poses modality-specific challenges: discriminative cues may be embedded in long-term trends, feature relevance can vary over time and across channels, and common perturbation-based explainers may produce out-of-distribution counterfactuals that compromise faithfulness (Liu et al., 2024c; Ye & Keogh, 2009; Szegedy et al., 2013). To systematically assess these issues, Ismail et al. (2020) benchmark saliency methods for multivariate time series on synthetic data with ground-truth temporal–feature importance, finding that many explainers are unreliable across RNN/TCN/Transformer backbones; they also propose Temporal Saliency Rescaling (TSR) to improve localization by first identifying salient time steps and then refining per-feature attributions within them. More recent explainers explicitly target *in-distribution* and *pattern-coherent* rationales. TIMEX (Queen et al., 2023) encourages consistency between explainer and model latent representations, and TIMEX++ (Liu et al., 2024c) further generates label-preserving temporal patterns via an information-theoretic objective that minimizes mutual information with the original series while maximizing mutual information with the prediction. **Our work is complementary but security-focused:** rather than proposing another explainer, we study whether such modern, pattern-aware explainers remain reliable under *adaptive* adversaries. We show that an attacker can simultaneously force a target label and steer explanations toward a chosen *reference* rationale, enabling a realistic *cover-up* failure mode for explanation-based auditing.

**Adversarial attacks on time series models.**    Adversarial robustness has been extensively studied in vision/NLP (Goodfellow et al., 2014; Chakraborty et al., 2018; Goyal et al., 2023), and a growing body of work develops attacks tailored to time series classification and forecasting. Early approaches largely transplant image-inspired pipelines. For instance, Karim et al. (2020) adapt adversarial transformation networks to synthesize adversarial time series examples against multiple classifiers. Subsequent work incorporates time series structure and threat-model constraints. Ding et al. (2023) propose BlackTreeS, which identifies top-$K$ important temporal regions via tree-position search and then performs region-focused black-box gradient estimation for more imperceptible attacks. Gu et al. (2025) (SFAttack) further improves stealth by constraining perturbations to discriminative shapelets and to high-frequency components that are less perceptible to human inspection. For forecasting, Mode & Hoque (2020) adapt classical gradient-based attacks (e.g., FGSM/BIM) to regression settings and demonstrate the fragility of forecasting networks, while Liu et al. (2023) study multivariate settings and propose indirect attacks (perturbing correlated series) as well as sparse, stealthy perturbations affecting only a subset of related signals. In parallel, a smaller line examines *explainer* robustness via diagnostic perturbations. AMEE (Nguyen et al., 2024) evaluates and ranks explainers by injecting random Gaussian noise into the regions they identify as discriminative; more informative explanations should induce a larger drop in predictive accuracy when those regions are perturbed. **Key limitation across these lines:** existing time series attacks predominantly target predictions (without controlling post-attack explanations), while explainer robustness diagnostics typically rely on *non-adaptive* random perturbations rather than an adversary that optimizes against the explainer. **In contrast,** our setting is an *active* and *dual-target* manipulation problem: we optimize to (i) force the classifier to a designated label and (ii) steer the explainer toward a specified target (often a reference rationale for auditing), exposing a threat that prediction-only robustness tests and random-noise diagnostics do not capture.

**Explainer robustness in other domains.**    Compared to time series, explainer robustness has been more extensively studied in vision and NLP. In image classification, Ghorbani et al. (2019) show that saliency maps from popular gradient-based

methods can be drastically altered by imperceptible perturbations while preserving the predicted label, revealing the fragility of attribution-based explanations. Zhang et al. (2020) further demonstrate black-box attacks on interpretable systems, including attacks that selectively fool explanations or jointly mislead predictions and explanation reports. In NLP, Ivankay et al. (2022) propose TextExplanationFooler (TEF), using semantics-preserving textual perturbations to significantly change token-level attributions while keeping classifier outputs stable. Beyond these, attacks on explainers have also been explored in the graph domain (Fan et al., 2023; Li et al., 2024). While these results establish that explanation manipulation is possible in other modalities, transferring the same capability to time series is not straightforward. time series auditing often relies on *temporally coherent, pattern-like* rationales, and our analysis highlights a modality-specific *high-dimensional paradox*: although the $T \times D$ input space offers a large surface for flipping predictions, unconstrained optimization tends to yield dense, temporally incoherent updates that are difficult to steer toward a *target* explanation (Section 4.1). **Our contribution is time series specific:** we instantiate a time–frequency structured dual-target attack that achieves pattern-level control, enabling targeted label manipulation while matching a reference rationale for explanation-based auditing.

## B. Proof of Theorem 4.1

We prove Eq. (1)–(2). Recall $A(\mathbf{X}) = \mathcal{H}^E(\mathbf{X}) \in \mathbb{R}^d$ and the dense $\ell_\infty$ step $\delta = -\varepsilon \operatorname{sign}(g_c)$ with $\tilde{\mathbf{X}} = \mathbf{X} + \delta$. Here $g_c$ denotes the classification gradient used by the attacker, i.e.,

$$g_c := \nabla_{\mathbf{X}} \mathcal{L}_{\text{cls}}(f(\mathbf{X}), y_{\text{tgt}}),$$

and $\mathbb{E}[\cdot]$ is taken over $\mathbf{X}$ drawn from the evaluation distribution.

**Notation.** For a set $S \subseteq [d]$, define $\|v\|_{1,S} := \sum_{i \in S} |v_i|$. In our setting, $\Omega$ corresponds to the reference (clean) rationale window. We assume $\mathbf{A}' \in \{0,1\}^d$ is strictly supported on $\Omega$, i.e., $\|\mathbf{A}'\|_{1,\Omega^c} = 0$. For each coordinate $i$, let $A_i(\mathbf{X})$ denote the $i$-th attribution entry.

**Assumptions (formalized).** We make the following local regularity assumptions along the dense update direction. Let the directional derivative of $A_i$ at $\mathbf{X}$ along $\delta$ be

$$D_\delta A_i(\mathbf{X}) := \nabla A_i(\mathbf{X})^\top \delta.$$

**Assumption B.1** (Off-window directional sensitivity). There exist constants $\eta \in (0,1)$, $\gamma > 0$ and a set $\mathcal{U} \subseteq \Omega^c$ with $|\mathcal{U}| \geq \eta(d - |\Omega|)$ such that for all $i \in \mathcal{U}$,

$$\mathbb{E}\big[\big|\nabla A_i(\mathbf{X})^\top \operatorname{sign}(g_c)\big|\big] \geq \gamma. \tag{6}$$

For standard time series explainers, the attribution $A_i(\mathbf{X})$ is a complex, non-linear function of the input (Sundararajan et al., 2017; Crabbé & Van Der Schaar, 2021; Huang et al., 2026; Leung et al., 2023; Chuang et al., 2023; Queen et al., 2023; Liu et al., 2024c). Unless an explainer explicitly employs a hard-masking mechanism on $\Omega^c$, the gradient $\nabla A_i(\mathbf{X})$ is non-vanishing for the majority of off-window coordinates (Ghorbani et al., 2019; Dombrowski et al., 2019). Furthermore, as the classification gradient sign $\operatorname{sign}(g_c) \in \{\pm 1\}^d$ is a dense vector, the directional derivative $\nabla A_i(\mathbf{X})^\top \operatorname{sign}(g_c)$ would vanish only under exact, non-generic cancellation across many input dimensions. Such perfect cancellation is statistically improbable and unstable in the presence of even infinitesimal noise, ensuring that $\gamma$ remains bounded away from zero in practice.

**Assumption B.2** (Directional curvature bound under $\ell_\infty$). There exists $L > 0$ such that for all $i \in \mathcal{U}$ and all $t \in [0,1]$,

$$\big|\delta^\top \nabla^2 A_i(\mathbf{X} + t\delta) \delta\big| \leq L \|\delta\|_\infty^2. \tag{7}$$

**Assumption B.3** (Bounded clean background magnitude). There exists a constant $\beta_0 \geq 0$ such that for all $i \in \mathcal{U}$,

$$\mathbb{E}[|A_i(\mathbf{X})|] \leq \beta_0. \tag{8}$$

Assumption B.3 states that off-window attributions have a bounded baseline magnitude on clean inputs, consistent with the regime where explanations are concentrated on the salient window $\Omega$.

**Step 1: A coordinate-wise lower bound on $|A_i(\mathbf{X}+\delta)|$.** Fix any $i \in \mathcal{U}$ and define the scalar function $\phi_i(t) := A_i(\mathbf{X}+t\delta)$ for $t \in [0,1]$. By Taylor's theorem along the direction $\delta$, there exists $\xi_i \in (0,1)$ such that

$$A_i(\mathbf{X} + \delta) = A_i(\mathbf{X}) + \underbrace{\nabla A_i(\mathbf{X})^\top \delta}_{=:\, b_i} + \underbrace{\frac{1}{2} \delta^\top \nabla^2 A_i(\mathbf{X} + \xi_i\delta)\, \delta}_{=:\, r_i}. \tag{9}$$

By Assumption B.2,

$$|r_i| \leq \frac{L}{2} \|\delta\|_\infty^2. \tag{10}$$

Using the reverse triangle inequality and then the triangle inequality,

$$|A_i(\mathbf{X} + \delta)| = |b_i + (A_i(\mathbf{X}) + r_i)| \geq |b_i| - |A_i(\mathbf{X}) + r_i| \geq |b_i| - |A_i(\mathbf{X})| - |r_i|.$$

Substituting $\delta = -\varepsilon \operatorname{sign}(g_c)$ gives

$$|b_i| = \varepsilon \left|\nabla A_i(\mathbf{X})^\top \operatorname{sign}(g_c)\right|, \qquad \|\delta\|_\infty = \varepsilon.$$

Taking expectation and using Eq. (10), Assumption B.1, and Assumption B.3, we obtain

$$\mathbb{E}[|A_i(\mathbf{X} + \delta)|] \geq \varepsilon\, \mathbb{E}\left[\left|\nabla A_i(\mathbf{X})^\top \operatorname{sign}(g_c)\right|\right] - \mathbb{E}[|A_i(\mathbf{X})|] - \mathbb{E}[|r_i|]$$

$$\geq \gamma\,\varepsilon - \beta_0 - \frac{L}{2}\,\varepsilon^2. \tag{11}$$

**Step 2: Sum over many off-window coordinates (no sign issue).** Since $|A_i(\mathbf{X}+\delta)| \geq 0$, we can safely drop nonnegative terms:

$$\mathbb{E}[\|A(\mathbf{X} + \delta)\|_{1,\Omega^c}] = \sum_{i \in \Omega^c} \mathbb{E}[|A_i(\mathbf{X} + \delta)|] \geq \sum_{i \in \mathcal{U}} \mathbb{E}[|A_i(\mathbf{X} + \delta)|]. \tag{12}$$

Combining Eq. (12) with Eq. (11) yields

$$\mathbb{E}[\|A(\mathbf{X} + \delta)\|_{1,\Omega^c}] \geq |\mathcal{U}| \left(\gamma\,\varepsilon - \beta_0 - \frac{L}{2}\,\varepsilon^2\right). \tag{13}$$

Using $|\mathcal{U}| \geq \eta(d - |\Omega|)$, we obtain

$$\mathbb{E}[\|A(\mathbf{X} + \delta)\|_{1,\Omega^c}] \geq \eta(d - |\Omega|) \left(\gamma\,\varepsilon - \beta_0 - \frac{L}{2}\,\varepsilon^2\right). \tag{14}$$

For any

$$\varepsilon \in \left[\frac{4\beta_0}{\gamma}, \frac{\gamma}{2L}\right], \tag{15}$$

The interval in (15) is non-empty iff $\frac{4\beta_0}{\gamma} \leq \frac{\gamma}{2L}$, i.e., $\beta_0 \leq \frac{\gamma^2}{8L}$. This condition is plausible in our setting since standard time series explainers typically concentrate attribution on the salient window, making the expected off-window baseline magnitude $\beta_0$ sufficiently small.

Based on Eq. 14,

$$\beta_0 \leq \frac{\gamma}{4}\varepsilon, \qquad \frac{L}{2}\varepsilon^2 \leq \frac{\gamma}{4}\varepsilon,$$

and hence

$$\gamma\varepsilon - \beta_0 - \frac{L}{2}\varepsilon^2 \geq \frac{\gamma}{2}\varepsilon.$$

Let

$$c := \frac{\eta\gamma}{2}. \tag{16}$$

Then Eq. (14) implies Eq. (1):

$$\mathbb{E}[\|A(\mathbf{X} + \delta)\|_{1,\Omega^c}] \geq c\varepsilon\,(d - |\Omega|).$$

**Step 3: Lower bound the mismatch to the target hard mask.** Since $\mathbf{A}'$ is supported on $\Omega$, we have $(\mathbf{A}')_i = 0$ for all $i \in \Omega^c$. Therefore,

$$
\begin{aligned}
\|A(\mathbf{X} + \delta) - \mathbf{A}'\|_1 &= \sum_{i \in \Omega} |A_i(\mathbf{X} + \delta) - 1| + \sum_{i \in \Omega^c} |A_i(\mathbf{X} + \delta) - 0| \\
&\geq \sum_{i \in \Omega^c} |A_i(\mathbf{X} + \delta)| = \|A(\mathbf{X} + \delta)\|_{1,\Omega^c}.
\end{aligned}
\tag{17}
$$

Taking expectation and combining with Eq. (1) gives

$$
\mathbb{E}[\|A(\mathbf{X} + \delta) - \mathbf{A}'\|_1] \geq \mathbb{E}[\|A(\mathbf{X} + \delta)\|_{1,\Omega^c}] \geq c\,\varepsilon\,(d - |\Omega|),
$$

which is Eq. (2). $\qquad\qquad\square$

## C. TSEF Algorithm

---

**Algorithm 1** TimeSeriesExplanationFooler (TSEF)

---

**Require:** Input series $\mathbf{X}$; Target label $y'$ and attribution $\mathbf{A}'$; Predictor $f$ and Explainer $\mathcal{H}^E$; Budget $\epsilon$; Iterations $I, K_t, K_f$;
    Learning rates $\eta_t, \eta_f$;
**Ensure:** Adversarial time series $\mathbf{X}^{\text{adv}}$.
1:  $\mathbf{X}^{\min}, \mathbf{X}^{\max} \leftarrow \min(\mathbf{X}), \max(\mathbf{X})$
2:  $\epsilon' \leftarrow \epsilon \cdot (\mathbf{X}^{\max} - \mathbf{X}^{\min})$ {Instance-wise perturbation budget}
3:  $\mathbf{X}_1 \leftarrow \mathbf{X}$
4:  **for** $i = 1, \ldots, I - 1$ **do**
5:     Reset $\mathbf{M}_t$ and $\Theta_f$ {re-identify vulnerable patterns on current $\mathbf{X}_i$}
6:     **// Stage 1: Optimize Temporal Mask ($\mathbf{M}_t$)**
7:     **for** $k = 1, \ldots, K_t$ **do**
8:         Sample binary mask $\tilde{\mathbf{M}}_t \sim \text{Concrete}(\mathbf{M}_t)$ {Gumbel-Sigmoid relaxation}
9:         $\mathbf{X}' \leftarrow \mathbf{X}_i \odot (1 - \tilde{\mathbf{M}}_t)$
10:       Compute loss $\mathcal{L}_t(\mathbf{M}_t)$
11:       $\mathbf{M}_t \leftarrow \Pi_{[0,1]} (\mathbf{M}_t - \eta_t \cdot \text{sign}(\nabla_{\mathbf{M}_t} \mathcal{L}_t))$ {Amplitude-invariant update}
12:     **end for**
13:     **// Stage 2: Optimize Frequency Perturbation Filter**
14:     **for** $k = 1, \ldots, K_f$ **do**
15:         Sample mask $\tilde{\mathbf{M}}_t$ using optimized $\mathbf{M}_t$
16:         Get segment spectrum: $\widehat{W} \leftarrow \mathcal{F}(\tilde{\mathbf{M}}_t \odot \mathbf{X}_i)$
17:         Compute unit perturbation: $\Delta\mathbf{X}_{\text{base}} \leftarrow \mathcal{F}^{-1}(\widehat{W} \odot \tanh(\Theta_f))$
18:         Compute scaling: $\alpha_{\text{freq}} \leftarrow \gamma \frac{\epsilon'}{\|\Delta\mathbf{X}_{\text{base}}\|_\infty + \tau}$
19:         Construct filter: $\mathbf{M}_f \leftarrow \Pi_{[0,2]}(1 + \alpha_{\text{freq}} \tanh(\Theta_f))$
20:         Reconstruct: $\tilde{\mathbf{X}} \leftarrow \mathbf{X}_i \odot (1 - \tilde{\mathbf{M}}_t) + \mathcal{F}^{-1}(\widehat{W} \odot \mathbf{M}_f)$
21:         Compute attack loss $J_{\text{atk}}(\tilde{\mathbf{X}})$
22:         $\Theta_f \leftarrow \Theta_f - \eta_f \cdot \text{sign}(\nabla_{\Theta_f} J_{\text{atk}})$ {Energy-invariant update}
23:     **end for**
24:     Clip the perturbed input to satisfy the norm constraint: $\left\|\tilde{\mathbf{X}} - \mathbf{X}\right\|_\infty \leq \epsilon'$
25:     Clip $\tilde{\mathbf{X}}$ element-wise to the range $[\mathbf{X}^{\min}, \mathbf{X}^{\max}]$
26:     $\mathbf{X}_{i+1} \leftarrow \tilde{\mathbf{X}}$
27: **end for**
28: $\mathbf{X}^{\text{adv}} \leftarrow \mathbf{X}_I$
29: **return** $\mathbf{X}^{\text{adv}}$

---

Algorithm 1 details the implementation of **TSEF** introduced in Section 4.2. Following the high-dimensional paradox (Section 4.1), TSEF turns dual-target feasibility into *structured controllability* by decoupling the attack into *when* to attack

and *how* to attack. Concretely, TSEF alternates two stages under an $\ell_\infty$ budget: (i) **Temporal Vulnerability Mask** learns a sparse, connected support $\mathbf{M}_t$ that identifies vulnerable temporal patterns for the *joint* objective (target prediction and reference explanation), using a Concrete relaxation with a straight-through estimator for discrete-like pattern selection; (ii) **Frequency Perturbation Filter** then performs a constrained spectral edit within this support, modifying trend/periodicity while keeping the signal largely intact. Both stages use sign-based updates to reduce sensitivity to raw amplitude/energy scales, and we adaptively rescale the frequency edit to satisfy the instance-wise $\ell_\infty$ constraint. The final adversarial series is obtained by reconstructing the edited segment via $\mathcal{F}^{-1}$ and reinserting it into the unmasked context. We convert the normalized budget $\epsilon$ into an instance-wise range-aware bound $\epsilon' = \epsilon(\mathbf{X}^{\max} - \mathbf{X}^{\min})$ to make perturbations comparable across signals with different amplitudes.

# D. Properties of the TSEF Optimization (Section 4.2)

## D.1. Tractable upper bound on mutual information

**Proposition.** Let $(X, M)$ be random variables with conditional distribution $P(M \mid X)$ and marginal $P(M)$. Let $Q(M)$ be any fixed prior independent of $X$.

Then

$$I(X; M) \;\leq\; \mathbb{E}_X\big[\mathrm{KL}\big(P(M \mid X) \,\|\, Q(M)\big)\big].$$

*Proof.* By definition,

$$I(X; M) = \mathbb{E}_X\big[\mathrm{KL}\big(P(M \mid X) \,\|\, P(M)\big)\big] = \mathbb{E}_{X,M}\left[\log \frac{p(M \mid X)}{p(M)}\right],$$

where $p(\cdot)$ denotes a density or mass function. Similarly,

$$\mathbb{E}_X\big[\mathrm{KL}\big(P(M \mid X) \,\|\, Q(M)\big)\big] = \mathbb{E}_{X,M}\left[\log \frac{p(M \mid X)}{q(M)}\right].$$

Using the decomposition

$$\log \frac{p(M \mid X)}{q(M)} = \log \frac{p(M \mid X)}{p(M)} + \log \frac{p(M)}{q(M)},$$

and taking expectations yields the identity

$$\mathbb{E}_X\big[\mathrm{KL}\big(P(M \mid X) \,\|\, Q(M)\big)\big] = I(X; M) + \mathrm{KL}\big(P(M) \,\|\, Q(M)\big).$$

Since $\mathrm{KL}(P(M)\|Q(M)) \geq 0$, we conclude $I(X; M) \leq \mathbb{E}_X[\mathrm{KL}(P(M \mid X)\|Q(M))]$. $\qquad\square$

## D.2. Amplitude-invariant Step of the Temporal Mask

We show that the gradient-based update for the temporal mask selects vulnerable time–channels by testing the *directional alignment* between the input and the loss gradient, and that this selection is invariant to the raw amplitude of $\mathbf{X}$.

**Proposition D.1** (Amplitude-invariant optimization). *Let $\mathbf{X} \in \mathbb{R}^{T \times D}$ be the input. Let $\mathbf{M}_t \in [0,1]^{T \times D}$ denote the continuous mask probabilities (which parameterize the Gumbel-Softmax distribution). For the purpose of gradient analysis, we consider the **deterministic relaxation** $\mathbf{X}' = \mathbf{X} \odot (1 - \mathbf{M}_t)$. The gradient of any differentiable loss $\mathcal{L}_t$ with respect to the mask $\mathbf{M}_t$ is:*

$$\frac{\partial \mathcal{L}_t}{\partial \mathbf{M}_t[t, d]} \;=\; -\mathbf{X}[t, d] \cdot \frac{\partial \mathcal{L}_t}{\partial \mathbf{X}'[t, d]}. \tag{18}$$

*Consequently, the sign of the gradient depends only on the **directional correlation** between the input feature $\mathbf{X}[t, d]$ and the loss sensitivity $\nabla_{\mathbf{X}'}\mathcal{L}_t$, independent of the magnitude $|\mathbf{X}[t, d]|$.*

*Furthermore, under the parameterization $\mathbf{M}_t(\Theta_t) = \sigma(\Theta_t)$, the update of the logits $\Theta_t$ inherits this invariance:*

$$\mathrm{sign}\left(\frac{\partial \mathcal{L}_t}{\partial \Theta_t[t, d]}\right) \;=\; -\,\mathrm{sign}(\mathbf{X}[t, d]) \cdot \mathrm{sign}\left(\frac{\partial \mathcal{L}_t}{\partial \mathbf{X}'[t, d]}\right). \tag{19}$$

*Thus, the update prioritizes coordinates where the fixed perturbation direction $(-\mathbf{X})$ aligns with the local descent direction, regardless of signal amplitude.*

*Proof.* **Gradient Derivation.** By the element-wise masking relation $\mathbf{X}'[t,d] = \mathbf{X}[t,d] \cdot (1 - \mathbf{M}_t[t,d])$, a partial derivative calculation via the chain rule yields:

$$\frac{\partial \mathcal{L}_t}{\partial \mathbf{M}_t[t,d]} = \frac{\partial \mathcal{L}_t}{\partial \mathbf{X}'[t,d]} \cdot \frac{\partial \mathbf{X}'[t,d]}{\partial \mathbf{M}_t[t,d]} = \frac{\partial \mathcal{L}_t}{\partial \mathbf{X}'[t,d]} \cdot (-\mathbf{X}[t,d]).$$

This confirms Eq. (18).

**Amplitude Invariance.** From Eq. (18), the magnitude of the gradient scales linearly with the input amplitude:

$$\left| \frac{\partial \mathcal{L}_t}{\partial \mathbf{M}_t[t,d]} \right| \propto |\mathbf{X}[t,d]|.$$

Standard gradient descent would be biased toward high-amplitude regions. However, taking the *sign* removes this factor:

$$\text{sign}\left( \frac{\partial \mathcal{L}_t}{\partial \mathbf{M}_t[t,d]} \right) = \text{sign}(-\mathbf{X}[t,d]) \cdot \text{sign}(\nabla_{\mathbf{X}'} \mathcal{L}_t).$$

This direction relies solely on whether suppressing the feature (moving along $-\mathbf{X}$) reduces the loss, decoupling the selection from the raw scale $|\mathbf{X}|$.

**Invariance for $\Theta_t$.** In practice, we implement this update in the logit space $\Theta_t$ (with $\mathbf{M}_t = \sigma(\Theta_t)$ where $\sigma(\cdot)$ is the sigmoid function) for stable parameterization; we write the update in terms of $\mathbf{M}_t$ in the main text for notational simplicity. By the chain rule:

$$\frac{\partial \mathcal{L}_t}{\partial \Theta_t} = \frac{\partial \mathcal{L}_t}{\partial \mathbf{M}_t} \cdot \frac{\partial \mathbf{M}_t}{\partial \Theta_t} = \frac{\partial \mathcal{L}_t}{\partial \mathbf{M}_t} \cdot \underbrace{\left[ \sigma(\Theta_t)(1 - \sigma(\Theta_t)) \right]}_{>0}.$$

Since the derivative of the sigmoid function is strictly positive, the sign is preserved:

$$\text{sign}\left( \frac{\partial \mathcal{L}_t}{\partial \Theta_t} \right) = \text{sign}\left( \frac{\partial \mathcal{L}_t}{\partial \mathbf{M}_t} \right).$$

Therefore, the sign-based update on logits $\Theta_t$ is mathematically equivalent to the amplitude-invariant selection derived for $\mathbf{M}_t$. $\square$

**Interpretation.** Because increasing $\mathbf{M}_t$ (masking) moves the input along the fixed direction $-\mathbf{X}$, the "vulnerable" positions are precisely those where this direction aligns with the negative gradient of the loss. Using the sign update ensures that large-amplitude artifacts do not dominate learning: coordinates are selected by *sensitivity* (gradient direction) rather than raw signal size.

### D.3. Energy-invariant Step of the Frequency Filter

**Proposition D.2** (Energy-invariant optimization). *Let $W = \mathbf{M}_t^* \odot \mathbf{X}$, and let $\widehat{W} = \mathcal{F}(W) \in \mathbb{C}^{K \times D}$ denote its discrete Fourier transform. Consider a real-valued multiplicative filter $\mathbf{M}_f \in [0,2]^{K \times D}$ applied as $\widehat{W}' = \widehat{W} \odot \mathbf{M}_f$, followed by reconstruction $\widetilde{W} = \mathcal{F}^{-1}(\widehat{W}')$ and loss evaluation $J_{\text{atk}}(\widetilde{W})$. Let the spectral energy of a bin be $E[k,d] := |\widehat{W}[k,d]|^2$.*

*The gradient of the loss with respect to the filter $\mathbf{M}_f$ is given by:*

$$\frac{\partial J_{\text{atk}}}{\partial \mathbf{M}_f[k,d]} = \Re\left( \overline{\widehat{W}[k,d]} \cdot \frac{\partial J_{\text{atk}}}{\partial \widehat{W}'[k,d]} \right), \tag{20}$$

*where $\Re(\cdot)$ denotes the real part and the overline denotes complex conjugation. Consequently, the sign-based update direction is invariant to the spectral amplitude $|\widehat{W}[k,d]|$ (and thus invariant to the energy $E[k,d]$).*

*Furthermore, under the parameterization $\mathbf{M}_f(\Theta_f) = \Pi_{[0,2]}(1 + \alpha_{\text{freq}} \tanh(\Theta_f))$, the update of the logits $\Theta_f$ inherits this invariance:*

$$\text{sign}\left( \frac{\partial J_{\text{atk}}}{\partial \Theta_f[k,d]} \right) = \text{sign}\left( \Re\left( \overline{\widehat{W}[k,d]} \frac{\partial J_{atk}}{\partial \widehat{W}'[k,d]} \right) \right). \tag{21}$$

*Proof.* **Gradient Derivation.** Since $\mathbf{M}_f[k, d] \in \mathbb{R}$ is a scalar scaling factor for the complex variable $\widehat{W}[k, d]$, a perturbation $\Delta\mathbf{M}_f[k, d]$ induces

$$\Delta\widehat{W}'[k, d] \;=\; \widehat{W}[k, d] \cdot \Delta\mathbf{M}_f[k, d].$$

Using standard $\mathbb{CR}$-calculus (Wirtinger calculus) rules (Kreutz-Delgado, 2009), the first-order variation of the real-valued function $J_{\mathrm{atk}}$ can be written as

$$\Delta J_{\mathrm{atk}} \;\approx\; \sum_{k,d} \Re\left( \overline{\frac{\partial J_{\mathrm{atk}}}{\partial \widehat{W}'[k, d]}} \cdot \Delta\widehat{W}'[k, d] \right),$$

where the complex conjugation appears on the gradient term by the real-valued first-order variation formula. Substituting $\Delta\widehat{W}'[k, d] = \widehat{W}[k, d]\Delta\mathbf{M}_f[k, d]$ yields

$$\Delta J_{\mathrm{atk}} \;\approx\; \sum_{k,d} \Re\left( \overline{\frac{\partial J_{\mathrm{atk}}}{\partial \widehat{W}'[k, d]}} \cdot \widehat{W}[k, d] \right) \Delta\mathbf{M}_f[k, d].$$

Therefore,

$$\frac{\partial J_{\mathrm{atk}}}{\partial \mathbf{M}_f[k, d]} \;=\; \Re\left( \overline{\frac{\partial J_{\mathrm{atk}}}{\partial \widehat{W}'[k, d]}} \cdot \widehat{W}[k, d] \right) \;=\; \Re\left( \overline{\widehat{W}[k, d]} \cdot \frac{\partial J_{\mathrm{atk}}}{\partial \widehat{W}'[k, d]} \right),$$

where the last equality uses $\Re(\overline{A}B) = \Re(A\overline{B})$. This confirms Eq. (20).

**Energy Invariance.** From Eq. (20), we observe that the gradient magnitude scales linearly with the input spectral amplitude:

$$\left| \frac{\partial J_{\mathrm{atk}}}{\partial \mathbf{M}_f[k, d]} \right| \propto |\widehat{W}[k, d]| = \sqrt{E[k, d]}.$$

Consider a scaling of the input spectrum $\widehat{W} \mapsto c \cdot \widehat{W}$ for some $c > 0$. The gradient scales by $c$, but its *sign* remains unchanged:

$$\mathrm{sign}(c \cdot \nabla) = \mathrm{sign}(\nabla).$$

Thus, the sign update removes the dependence on the magnitude $|\widehat{W}|$, driven purely by the directional alignment (phase correlation) between the signal $\widehat{W}$ and the error sensitivity $\nabla_{\widehat{W}'} J$.

**Invariance for $\Theta_f$.** In TSEF, we update the logits $\Theta_f$ where $\mathbf{M}_f = 1 + \alpha_{\mathrm{freq}} \tanh(\Theta_f)$ (ignoring clipping for gradient direction). In implementation, we treat $\alpha_{\mathrm{freq}}$ as a normalization constant and detach it from the computation graph when updating $\Theta_f$; hence $\partial\alpha_{\mathrm{freq}}/\partial\Theta_f = 0$. By the chain rule:

$$\frac{\partial J_{\mathrm{atk}}}{\partial \Theta_f} \;=\; \frac{\partial J_{\mathrm{atk}}}{\partial \mathbf{M}_f} \cdot \frac{\partial \mathbf{M}_f}{\partial \Theta_f} \;=\; \frac{\partial J_{\mathrm{atk}}}{\partial \mathbf{M}_f} \cdot \underbrace{\left[ \alpha_{\mathrm{freq}} \cdot (1 - \tanh^2(\Theta_f)) \right]}_{>0}.$$

Since $\alpha_{\mathrm{freq}} > 0$ and $(1 - \tanh^2) > 0$, the partial derivative $\frac{\partial \mathbf{M}_f}{\partial \Theta_f}$ is a strictly positive scalar. Therefore,

$$\mathrm{sign}\left( \frac{\partial J_{\mathrm{atk}}}{\partial \Theta_f} \right) \;=\; \mathrm{sign}\left( \frac{\partial J_{\mathrm{atk}}}{\partial \mathbf{M}_f} \right).$$

This proves that the energy-invariant property of the filter-space gradient directly governs the update direction of the parameter $\Theta_f$. □

## E. Description of Datasets

We follow the dataset construction and preprocessing protocols of (Queen et al., 2023; Liu et al., 2024c) and use both synthetic and real-world benchmarks. Below we summarize the datasets used in our experiments and how ground-truth explanations are defined when available.

## E.1. Synthetic datasets

We adopt three synthetic datasets with known ground-truth explanations to study how well attacks and explainers can identify the underlying predictive signal. Following (Queen et al., 2023; Liu et al., 2024c), all synthetic time series are built on top of a non-autoregressive moving average (NARMA) noise process, into which class-specific patterns are inserted. The design follows standard recommendations for synthetic benchmarks that avoid shortcut solutions and require the model to detect precise temporal patterns rather than simple heuristics. For each synthetic dataset, we generate 5,000 training samples, 1,000 test samples, and 100 validation samples.

**SEQCOMB-UV.** SEQCOMB-UV is a univariate dataset where the predictive signal is defined by the presence and combination of two subsequence "shapes": increasing (I) and decreasing (D) trends. For each time series, two non-overlapping subsequences of length 10–20 time steps are first selected. Depending on the class label, either an increasing or decreasing pattern is inserted into each subsequence by superimposing a sinusoidal trend with a randomly chosen wavelength. The four classes are: class 0 (null, no structured subsequence), class 1 (I, I), class 2 (D, D), and class 3 (I, D). Thus, correct classification requires detecting both subsequences and their trend directions. Ground-truth explanations are defined as the time indices of the I and D subsequences that determine the class.

**SEQCOMB-MV.** SEQCOMB-MV is the multivariate counterpart of SEQCOMB-UV. The overall construction and class structure are identical, but the I and D subsequences are distributed across different sensors. For each sample, the two subsequences are assigned to randomly selected channels. Ground-truth explanations mark the predictive subsequences on their respective sensors, i.e., both the time points at which the causal signal occurs and the channels on which they appear.

**LOWVAR.** In LOWVAR, the predictive signal is defined by a low-variance region embedded in a multivariate time series. As in the SeqComb datasets, a random subsequence is chosen in each sample, and within this segment the NARMA background is replaced by Gaussian noise with low variance. Classes are distinguished by the mean of this low-variance Gaussian noise and by the sensor on which the subsequence occurs: for class 0 the subsequence has mean $-1.5$ on sensor 0; for class 1, mean $1.5$ on sensor 0; for class 2, mean $-1.5$ on sensor 1; and for class 3, mean $1.5$ on sensor 1. Unlike anomaly-based datasets, the predictive region here is not obviously different in amplitude from the background, making it a more challenging explanation task. Ground-truth explanations correspond to the low-variance subsequences on the appropriate sensor.

## E.2. Real-world datasets

We also evaluate on three real-world time series classification benchmarks released by (Queen et al., 2023): ECG for arrhythmia detection, Epilepsy for seizure detection, and PAM for human activity recognition. We briefly summarize each dataset and how we use it.

**ECG (MIT-BIH).** The ECG dataset is derived from the MIT-BIH Arrhythmia database (Moody & Mark, 2001), which contains ECG recordings from 47 subjects sampled at 360 Hz. Following (Queen et al., 2023), the raw recordings are windowed into segments of length 360 time steps, producing 92,511 labeled beats. Two cardiologists independently annotated each beat; we adopt three labels for classification: normal (N), left bundle branch block (L), and right bundle branch block (R). Clinical knowledge indicates that L and R diagnoses primarily depend on the QRS complex, so the QRS interval is taken as the ground-truth explanatory region for L and R classes. For N, no specific explanatory segment is assumed.

**Epilepsy.** The Epilepsy dataset (Andrzejak et al., 2001) consists of single-channel EEG recordings from 500 subjects. For each subject, brain activity is recorded for 23.6 seconds and sampled at 178 Hz. The continuous signals are then partitioned and shuffled into 11,500 non-overlapping 1-second segments to weaken subject-specific correlations. The original annotations contain five categories: eyes open, eyes closed, EEG from a healthy region, EEG from a tumor region, and seizure activity. Following Queen et al. (2023), we merge the first four categories into a single "non-seizure" class and treat seizure vs. non-seizure as a binary classification problem. No ground-truth explanatory labels are provided for this dataset; we use it only for evaluating prediction attacks and explanation manipulation in the absence of human-annotated rationales.

**PAM.** The PAMAP (PAM) dataset (Reiss & Stricker, 2012) is a human activity recognition benchmark with wearable inertial measurement units. It contains continuous recordings of daily activities from 9 subjects measured by multiple sensors. Following Queen et al. (2023), we discard the ninth subject due to the short recording length, segment the remaining multivariate time series into windows of 600 time steps with 50% overlap, and remove rare classes with fewer than 500 samples. The resulting dataset has 5,333 segments, each with 600 time steps and 17 sensor channels, labeled into 8 activity

*Table 4.* Summary of time series datasets used in our experiments. For each dataset, we report the number of samples, sequence length, dimensionality, and number of classes. Synthetic datasets and ECG come with ground-truth explanation masks as described above, whereas PAM and Epilepsy are used without human-annotated explanations.

| DATASET | # SAMPLES | LENGTH | DIM. | # CLASSES | TASK |
|---|---|---|---|---|---|
| SEQCOMB-UV | 6,100 | 200 | 1 | 4 | MULTICLASS CLS. |
| SEQCOMB-MV | 6,100 | 200 | 4 | 4 | MULTICLASS CLS. |
| LOWVAR | 6,100 | 200 | 2 | 4 | MULTICLASS CLS. |
| ECG | 92,511 | 360 | 1 | 5 | ECG CLS. |
| PAM | 5,333 | 600 | 17 | 8 | ACTIVITY RECOG. |
| EPILEPSY | 11,500 | 178 | 1 | 2 | EEG CLS. |

classes (e.g., walking, sitting). The class distribution is approximately balanced after filtering. As with Epilepsy, PAM does not come with ground-truth explanation masks; we use it to study how attacks transfer to realistic multivariate sensor data.

**Evaluation setup.** We evaluate attacks on all correctly classified test samples. For the Epilepsy dataset, we evaluate on correctly classified seizure samples targeting non-seizure, as seizure detection represents the clinically critical minority class.

**Ground-truth explanation masks.** For the synthetic datasets (SEQCOMB-UV, SEQCOMB-MV, LOWVAR), ground-truth explanation masks are available by construction and mark the exact time indices (and, for multivariate data, channels) where the class-defining subsequences occur. Specifically, masks are 1 on the I/D subsequences in SEQCOMB-UV and SEQCOMB-MV, and on the low-variance subsequence in LOWVAR, and 0 elsewhere. For the ECG dataset, ground-truth explanations are defined on top of the cardiologist annotations: for L and R beats, we mark the QRS interval as the explanatory region, while normal beats have no designated explanatory segment. Epilepsy and PAM do not provide human-annotated rationales, and we therefore treat them as datasets without ground-truth explanations and only evaluate attacks in terms of their effect on predictions and explainer outputs.

# F. Hyperparameter Settings

**PGD.** For PGD, we perform targeted attacks by minimizing the targeted cross-entropy loss under an $L_\infty$ perturbation budget. We select PGD hyperparameters on a validation set via a small grid search over the number of iterations and step size: iterations $\in \{100, 200, 300\}$ and step sizes $\in \{1.0, 0.1, 0.05, 0.01, 0.001\}$.

**ADV$^2$.** We implement an ADV$^2$-style joint baseline adapted from (Zhang et al., 2020) by optimizing a weighted objective $\mathcal{L} = \lambda_{\text{cls}}\mathcal{L}_{\text{cls}} + \lambda_{\text{exp}}\mathcal{L}_{\text{exp}}$, where $\mathcal{L}_{\text{cls}}$ is the targeted classification loss and $\mathcal{L}_{\text{exp}}$ measures the discrepancy between the adversarial explanation and the reference explanation. For a fair comparison, we use the same time-series explanation-alignment loss (i.e., the same target definition and distance metric) as in TSEF. We select ADV$^2$ hyperparameters on a validation set via a grid search over the number of iterations and the step size: iterations $\in \{100, 200, 300\}$ and step sizes $\in \{1.0, 0.1, 0.05, 0.01, 0.001\}$. We also tune the weighting coefficients $\lambda_{\text{cls}}$ and $\lambda_{\text{exp}}$ on the validation set.

**BlackTreeS.** For BlackTreeS, we tune the step size in $\{1.0, 0.1, 0.01, 0.001\}$. We also tune the number of candidate most important time positions by the tree search, $\in \{T, 0.5T, 0.3T, 0.1T\}$, and the number of iterations in $\{100, 200, 300\}$.

**SFAttack.** For SFAttack, we tune the step size in $\{1.0, 0.1, 0.01, 0.001\}$ and the number of iterations in $\{100, 200, 300\}$. We additionally tune the cutoff frequency in $\{T/2, T/4, 1\}$ and the shapelet-length ratio in $\{0.9, 0.8, 0.7, 0.6, 0.5\}$.

**Local and global Gaussian perturbations.** Following AMEE (Nguyen et al., 2024), we first select the top-$k\%$ time steps by explainer saliency and then replace their values with Gaussian noise. For the *local* variant, we estimate per-time-step mean and variance $(\mu_t, \sigma_t^2)$ over the training set and sample $r_t \sim \mathcal{N}(\mu_t, \sigma_t^2)$. For the *global* variant, we instead estimate a global mean and variance $(\mu, \sigma^2)$ over all time steps and sample $r_t \sim \mathcal{N}(\mu, \sigma^2)$. We set $k = 10\%$ for all datasets unless otherwise noted.

**TSEF.** We first tune TSEF on the ECG validation set and fix the optimization schedule for all experiments: learning rates 1.0 for both $\mathbf{M}_t$ and $\mathbf{M}_f$, 100 outer iterations, and per-iteration inner-loop updates of 10 steps for $\mathbf{M}_t$ and 20 steps for $\mathbf{M}_f$, with $\lambda_{\text{spa}} = 1.0$ and $\lambda_{\text{con}} = 1.0$ for the temporal mask $\mathbf{M}_t$. We then reuse this schedule for all other datasets. For non-ECG datasets, we only tune the loss-weight hyperparameters (e.g., the trade-off between classification and explanation objectives)

on their validation sets, while keeping the learning rates, the inner/outer iteration counts, and $\lambda_{\mathrm{spa}}, \lambda_{\mathrm{con}}$ unchanged.

# G. Details of Metrics

We evaluate attacks along two axes: how well they mislead *explanations* and how well they mislead *predictions*. Below we detail the metrics used in each case.

## G.1. Misleading explanation performance

Following (Crabbé & Van Der Schaar, 2021), we treat explanation quality as a binary classification problem over time–feature indices. For a time series $X \in \mathbb{R}^{T \times D}$, an explainer produces a soft saliency mask $M \in [0,1]^{T \times D}$, where larger values indicate higher attributed importance. When ground-truth explanations are available, we denote them by a binary matrix $Q \in \{0,1\}^{T \times D}$, where $Q[t,d] = 1$ if the input feature $X[t,d]$ is truly salient and $Q[t,d] = 0$ otherwise.

Given a detection threshold $\tau \in (0,1)$, we obtain a binary estimate $\hat{Q}(\tau)$ from the soft mask $M$ via

$$\hat{Q}[t,d](\tau) = \begin{cases} 1 & \text{if } M[t,d] \geq \tau, \\ 0 & \text{otherwise.} \end{cases} \tag{22}$$

Let

$$A = \{(t,d) \in [1{:}T] \times [1{:}D] \mid Q[t,d] = 1\}, \tag{23}$$

$$\hat{A}(\tau) = \{(t,d) \in [1{:}T] \times [1{:}D] \mid \hat{Q}[t,d](\tau) = 1\} \tag{24}$$

be the sets of truly salient indices and indices selected by the explainer at threshold $\tau$, respectively. We then define the precision and recall at threshold $\tau$ as

$$\mathrm{P}(\tau) = \frac{|A \cap \hat{A}(\tau)|}{|\hat{A}(\tau)|}, \tag{25}$$

$$\mathrm{R}(\tau) = \frac{|A \cap \hat{A}(\tau)|}{|A|}, \tag{26}$$

whenever the denominators are non-zero. Varying $\tau$ traces out precision and recall curves

$$\mathrm{P} : (0,1) \to [0,1], \quad \tau \mapsto \mathrm{P}(\tau), \tag{27}$$

$$\mathrm{R} : (0,1) \to [0,1], \quad \tau \mapsto \mathrm{R}(\tau). \tag{28}$$

The *area under precision* (AUP) and *area under recall* (AUR) scores summarize these curves by integration over all thresholds:

$$\mathrm{AUP} = \int_0^1 \mathrm{P}(\tau)\, d\tau, \tag{29}$$

$$\mathrm{AUR} = \int_0^1 \mathrm{R}(\tau)\, d\tau. \tag{30}$$

Intuitively, AUP captures how precise the explainer is on average when it highlights features as salient, while AUR captures how completely it recovers the truly salient features across thresholds. In our experiments, we compute these quantities numerically by discretizing $\tau$ over a fine grid. When we evaluate attacks that aim to steer explanations toward a target mask (e.g., a synthetic ground-truth pattern or a chosen rationale), we instantiate $Q$ with this target mask and report AUP/AUR between the adversarial explanation and $Q$.

## G.2. Misleading classification performance

To quantify how effectively an attack alters model predictions, we report the *Attack Success Rate* (ASR). For a given attack setting (e.g., targeted or untargeted), an adversarial trial is deemed successful if the attacked classifier output satisfies the

*Table 5.* Attack results on the TimesFM 2.5 backbone. Best results on each dataset and metric are highlighted in bold.

| DATASET | METHOD | F1 ↑ | ASR ↑ | AUPRC ↑ | AUP ↑ | AUR ↑ |
|---|---|---|---|---|---|---|
| LOWVAR | PGD | **0.9996 ± 0.0003** | **0.9996 ± 0.0003** | 0.5021 ± 0.0073 | 0.3707 ± 0.0058 | 0.5476 ± 0.0054 |
| | ADV$^2$ | 0.9992 ± 0.0005 | 0.9992 ± 0.0004 | 0.7755 ± 0.0068 | 0.5649 ± 0.0058 | 0.7442 ± 0.0049 |
| | **TSEF (OURS)** | 0.9992 ± 0.0005 | 0.9992 ± 0.0004 | **0.7954 ± 0.0067** | **0.5772 ± 0.0058** | **0.7564 ± 0.0049** |
| SEQCOMB-UV | PGD | 0.6615 ± 0.0319 | 0.6620 ± 0.0357 | 0.2731 ± 0.0028 | 0.2718 ± 0.0028 | 0.6460 ± 0.0031 |
| | ADV$^2$ | 0.6169 ± 0.0260 | 0.6243 ± 0.0324 | 0.2834 ± 0.0030 | 0.2798 ± 0.0029 | 0.6471 ± 0.0031 |
| | **TSEF (OURS)** | **0.7900 ± 0.0403** | **0.7870 ± 0.0414** | **0.3258 ± 0.0033** | **0.3218 ± 0.0032** | **0.6520 ± 0.0031** |
| SEQCOMB-MV | PGD | 0.7391 ± 0.0208 | 0.7320 ± 0.0228 | 0.0853 ± 0.0015 | 0.1148 ± 0.0025 | 0.6287 ± 0.0038 |
| | ADV$^2$ | 0.7482 ± 0.0184 | 0.7418 ± 0.0208 | 0.1004 ± 0.0017 | 0.1246 ± 0.0026 | 0.6500 ± 0.0030 |
| | **TSEF (OURS)** | **0.8091 ± 0.0126** | **0.8022 ± 0.0159** | **0.1147 ± 0.0020** | **0.1326 ± 0.0028** | **0.6594 ± 0.0037** |

*Table 6.* Attack results on the CNN backbone. Best results on each dataset and metric are highlighted in bold.

| DATASET | METHOD | F1 ↑ | ASR ↑ | AUPRC ↑ | AUP ↑ | AUR ↑ |
|---|---|---|---|---|---|---|
| LOWVAR | PGD | **0.9932 ± 0.0008** | **0.9933 ± 0.0008** | 0.1840 ± 0.0039 | 0.1307 ± 0.0036 | 0.2729 ± 0.0034 |
| | ADV$^2$ | 0.9922 ± 0.0025 | 0.9921 ± 0.0025 | 0.4583 ± 0.0060 | 0.3654 ± 0.0054 | 0.5155 ± 0.0045 |
| | **TSEF (OURS)** | 0.9923 ± 0.0017 | 0.9923 ± 0.0017 | **0.5835 ± 0.0063** | **0.4697 ± 0.0056** | **0.6410 ± 0.0044** |
| SEQCOMB-UV | PGD | 0.5572 ± 0.0205 | 0.5301 ± 0.0222 | 0.1565 ± 0.0019 | 0.1332 ± 0.0018 | 0.6441 ± 0.0033 |
| | ADV$^2$ | **0.5574 ± 0.0201** | 0.5291 ± 0.0227 | 0.1697 ± 0.0020 | 0.1389 ± 0.0018 | 0.6515 ± 0.0033 |
| | **TSEF (OURS)** | 0.5560 ± 0.0194 | **0.5324 ± 0.0207** | **0.2033 ± 0.0024** | **0.1611 ± 0.0021** | **0.6649 ± 0.0032** |
| SEQCOMB-MV | PGD | **0.5394 ± 0.0082** | **0.5232 ± 0.0105** | 0.4290 ± 0.0039 | 0.5653 ± 0.0048 | 0.5229 ± 0.0041 |
| | ADV$^2$ | 0.5313 ± 0.0051 | 0.5128 ± 0.0061 | 0.4769 ± 0.0045 | 0.5833 ± 0.0050 | 0.5356 ± 0.0040 |
| | **TSEF (OURS)** | 0.5303 ± 0.0055 | 0.5209 ± 0.0061 | **0.5333 ± 0.0051** | **0.5965 ± 0.0051** | **0.5541 ± 0.0039** |

desired misclassification criterion (e.g., switches from the true label to a specified target label). Formally,

$$\text{ASR} = \frac{\#\text{ successful trials}}{\#\text{ total trials}}. \tag{31}$$

In our targeted experiments, a trial is counted as successful if the classifier prediction on the adversarial example equals the prescribed target label and differs from the original prediction on the clean input. ASR therefore measures the fraction of inputs on which the attack is able to enforce the desired misclassification under the specified perturbation budget.

While ASR captures whether the attack can hit the target label at least once, it does not assess how *cleanly* the adversarial predictions match the desired target distribution. We therefore also report the *target-class F1 score*. Let $y'$ denote the target label assigned to an input (in our setting, randomly chosen to differ from the true label), and let $\hat{y}_{\text{adv}}$ be the classifier prediction on the adversarial example. In practice, we report the target-class F1 score by treating $y'$ as the ground-truth label for each adversarial example and computing the standard F1 between $\hat{y}_{\text{adv}}$ and $y'$ (micro-averaged over attacked samples). This metric complements ASR: a high ASR with low target F1 indicates scattered, inconsistent targeting, whereas simultaneously high ASR and target-class F1 indicate that the attack reliably and coherently relabels inputs into the desired target classes.

## H. Additional Experiments

### H.1. Alternative classifier backbones.

Tables 5 and 6 report results on TimesFM 2.5 (Das et al., 2024) and CNN backbones, respectively, both paired with TIMEX++. On TimesFM 2.5, TSEF achieves the strongest explanation alignment across all datasets and improves target prediction success on SEQCOMB-UV, and SEQCOMB-MV. On LOWVAR, PGD obtains slightly higher F1/ASR, while TSEF remains comparable in target success and substantially improves explanation alignment over both PGD and ADV$^2$. On the CNN backbone, TSEF again consistently improves AUPRC, AUP, and AUR, while remaining competitive on F1/ASR. These results indicate that the observed prediction–explanation decoupling is not specific to the default Transformer classifier.

*Table 7.* Attack results in the original-explanation-preserving setting, where $\mathbf{A}' = \mathbf{A}$ and $\mathbf{A}$ denotes the original explanation of the clean input. Best results on each dataset and metric are highlighted in bold.

| DATASET | METHOD | F1 ↑ | ASR ↑ | AUPRC ↑ | AUP ↑ | AUR ↑ |
|---|---|---|---|---|---|---|
| **LOWVAR** | PGD | $0.8441 \pm 0.0400$ | $0.8568 \pm 0.0355$ | $0.2536 \pm 0.0043$ | $0.1915 \pm 0.0040$ | $0.3185 \pm 0.0037$ |
| | ADV$^2$ | $0.8465 \pm 0.0396$ | $0.8578 \pm 0.0355$ | $0.6245 \pm 0.0052$ | $0.4936 \pm 0.0046$ | $0.6233 \pm 0.0040$ |
| | **TSEF (OURS)** | $\mathbf{0.8703 \pm 0.0435}$ | $\mathbf{0.8839 \pm 0.0380}$ | $\mathbf{0.8342 \pm 0.0035}$ | $\mathbf{0.6938 \pm 0.0034}$ | $\mathbf{0.7700 \pm 0.0025}$ |
| **PAM** | PGD | $0.6041 \pm 0.0124$ | $0.5967 \pm 0.0142$ | $0.6874 \pm 0.0057$ | $0.6390 \pm 0.0036$ | $0.4443 \pm 0.0040$ |
| | ADV$^2$ | $0.6091 \pm 0.0170$ | $0.5967 \pm 0.0182$ | $0.7542 \pm 0.0050$ | $0.6719 \pm 0.0033$ | $0.4519 \pm 0.0039$ |
| | **TSEF (OURS)** | $\mathbf{0.6168 \pm 0.0182}$ | $\mathbf{0.6054 \pm 0.0185}$ | $\mathbf{0.8503 \pm 0.0037}$ | $\mathbf{0.6981 \pm 0.0031}$ | $\mathbf{0.4637 \pm 0.0038}$ |
| **EPILEPSY** | PGD | $0.1595 \pm 0.0111$ | $0.1908 \pm 0.0158$ | $0.6582 \pm 0.0061$ | $0.3454 \pm 0.0044$ | $0.8640 \pm 0.0021$ |
| | ADV$^2$ | $0.1595 \pm 0.0111$ | $0.1908 \pm 0.0158$ | $0.6783 \pm 0.0059$ | $0.3468 \pm 0.0043$ | $0.8670 \pm 0.0021$ |
| | **TSEF (OURS)** | $\mathbf{0.1610 \pm 0.0149}$ | $\mathbf{0.1937 \pm 0.0212}$ | $\mathbf{0.7661 \pm 0.0046}$ | $\mathbf{0.3549 \pm 0.0042}$ | $\mathbf{0.8895 \pm 0.0021}$ |

*Table 8.* Surrogate transfer attack results across explainers on **LowVar**. Best results for each transfer pair and metric are highlighted in bold.

| TRANSFER PAIR | METHOD | F1 ↑ | ASR ↑ | AUPRC ↑ | AUP ↑ | AUR ↑ |
|---|---|---|---|---|---|---|
| TIMEX++ → TIMEX | PGD | $0.6287 \pm 0.0391$ | $0.6550 \pm 0.0342$ | $0.2179 \pm 0.0042$ | $0.1481 \pm 0.0034$ | $0.3545 \pm 0.0035$ |
| | ADV$^2$ | $0.6226 \pm 0.0457$ | $0.6507 \pm 0.0395$ | $0.2746 \pm 0.0045$ | $0.1886 \pm 0.0036$ | $0.4129 \pm 0.0036$ |
| | **TSEF (OURS)** | $\mathbf{0.6514 \pm 0.0343}$ | $\mathbf{0.6686 \pm 0.0321}$ | $\mathbf{0.3606 \pm 0.0050}$ | $\mathbf{0.2477 \pm 0.0038}$ | $\mathbf{0.4976 \pm 0.0037}$ |
| TIMEX → TIMEX++ | PGD | $0.6583 \pm 0.0343$ | $0.6730 \pm 0.0301$ | $0.2564 \pm 0.0043$ | $0.1945 \pm 0.0040$ | $0.2983 \pm 0.0036$ |
| | ADV$^2$ | $0.6530 \pm 0.0384$ | $0.6684 \pm 0.0344$ | $0.3385 \pm 0.0049$ | $0.2559 \pm 0.0044$ | $0.3710 \pm 0.0041$ |
| | **TSEF (OURS)** | $\mathbf{0.6655 \pm 0.0317}$ | $\mathbf{0.6769 \pm 0.0285}$ | $\mathbf{0.4224 \pm 0.0054}$ | $\mathbf{0.3203 \pm 0.0047}$ | $\mathbf{0.4487 \pm 0.0045}$ |

## H.2. Original-explanation-preserving setting.

Table 7 evaluates a setting where $\mathbf{A}' = \mathbf{A}$, with $\mathbf{A}$ denoting the clean explanation of the original input. This setting is especially relevant when human-annotated rationales are unavailable and the clean explanation is used as the auditing reference. Across LOWVAR, PAM, and Epilepsy, TSEF outperforms PGD and ADV$^2$ on all five metrics. These results further support our main conclusion: targeted prediction manipulation can coexist with explanation preservation, and explanation stability alone is not a reliable indicator of decision trustworthiness.

## H.3. Surrogate transfer across explainers.

Table 8 reports surrogate-transfer results between independently trained Transformer+TIMEX and Transformer+TIMEX++ pairs on LOWVAR. This setting evaluates whether adversarial examples optimized on one predictor–explainer pair can transfer to an unseen predictor–explainer pair. TSEF consistently outperforms PGD and ADV$^2$ across both transfer directions and all five metrics. In particular, TSEF improves not only target prediction success (F1/ASR), but also explanation alignment (AUPRC/AUP/AUR), suggesting that the prediction–explanation vulnerability is not restricted to the exact predictor–explainer pair used for optimization. We emphasize that this is surrogate-transfer evidence rather than a fully black-box attack; developing fully black-box joint attacks remains an important future direction.

## H.4. Hyperparameter Sensitivity Analysis

Figure 4 examines the interplay between the two objectives in our dual-target attack. We sweep the classification weight $\lambda_{\text{cls}}$ while fixing the explanation weight at $\lambda_{\text{exp}} = 1$.

Across both LOWVAR and SEQCOMB-MV, increasing $\lambda_{\text{cls}}$ consistently improves prediction targeting (higher ASR and target F1), but induces a monotonic degradation in explanation alignment (lower AUPRC/AUP/AUR). This inverse relationship empirically confirms an inherent *prediction–explanation trade-off*: as optimization prioritizes misclassification, the resulting perturbations become progressively less compatible with matching the reference rationale required for cover-up.

The *strength* of this trade-off varies across datasets. On the more complex multivariate SEQCOMB-MV, explanation alignment degrades only mildly (e.g., AUPRC drops by $\sim 0.01$) even when $\lambda_{\text{cls}}$ increases by two orders of magnitude, indicating that TSEF can often raise attack confidence without a catastrophic loss of camouflage quality. Overall, $\lambda_{\text{cls}}$ acts

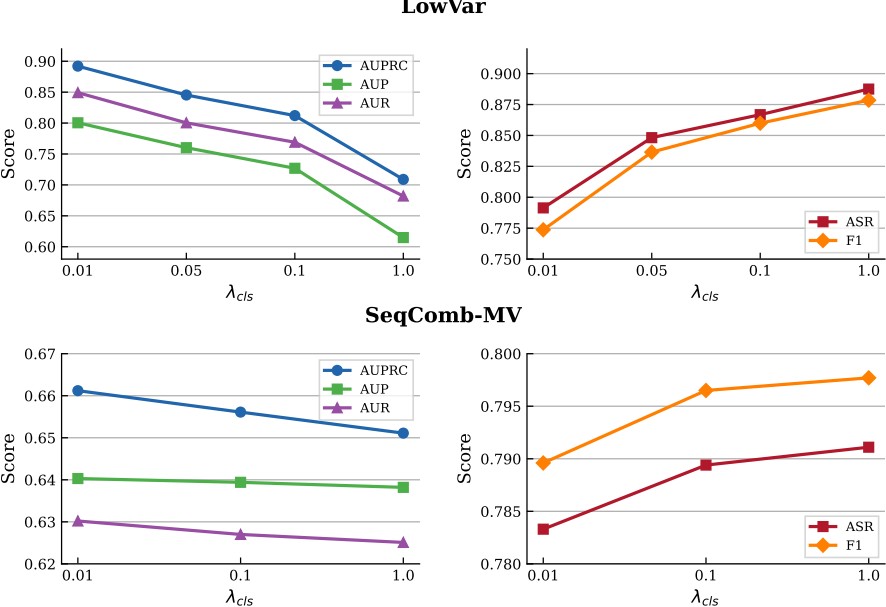

*Figure 4.* **Sensitivity to the classification weight** $\lambda_{\text{cls}}$**.** We sweep $\lambda_{\text{cls}}$ (with the explanation weight fixed, $\lambda_{\exp} = 1$) on LOWVAR (top) and SEQCOMB-MV (bottom). **Left:** explanation-targeting metrics (AUPRC/AUP/AUR) decrease as $\lambda_{\text{cls}}$ increases. **Right:** prediction-targeting metrics (ASR/F1) improve, revealing a clear trade-off between prediction success and explanation alignment.

*Table 9.* Ablation study on the LOWVAR and SEQCOMB-MV datasets. Best results are highlighted in bold.

| | | LOWVAR | | | | | SEQCOMB-MV | | | | |
|---|---|---|---|---|---|---|---|---|---|---|---|
| INTERPRETER | ATTACK | F1 ↑ | ASR ↑ | AUPRC ↑ | AUP ↑ | AUR ↑ | F1 ↑ | ASR ↑ | AUPRC ↑ | AUP ↑ | AUR ↑ |
| TIMEX++ | TSEF (w/o $\mathbf{M}_t$) | $0.743 \pm 0.030$ | $0.754 \pm 0.030$ | $0.796 \pm 0.004$ | $0.707 \pm 0.004$ | $0.702 \pm 0.003$ | $0.508 \pm 0.020$ | $0.491 \pm 0.021$ | $0.551 \pm 0.005$ | $0.621 \pm 0.005$ | $0.578 \pm 0.004$ |
| | TSEF (w/o $\mathbf{M}_f$) | $0.619 \pm 0.052$ | $0.657 \pm 0.043$ | $0.785 \pm 0.004$ | $0.697 \pm 0.004$ | $0.696 \pm 0.003$ | $0.686 \pm 0.016$ | $0.678 \pm 0.018$ | $0.632 \pm 0.006$ | $0.639 \pm 0.005$ | $0.614 \pm 0.003$ |
| | **TSEF (FULL)** | $\mathbf{0.837 \pm 0.046}$ | $\mathbf{0.848 \pm 0.042}$ | $\mathbf{0.845 \pm 0.004}$ | $\mathbf{0.760 \pm 0.004}$ | $\mathbf{0.800 \pm 0.003}$ | $\mathbf{0.795 \pm 0.012}$ | $\mathbf{0.789 \pm 0.015}$ | $\mathbf{0.661 \pm 0.006}$ | $\mathbf{0.641 \pm 0.005}$ | $\mathbf{0.631 \pm 0.003}$ |
| INTEGRATED GRADIENTS | TSEF (w/o $\mathbf{M}_t$) | $0.729 \pm 0.024$ | $0.739 \pm 0.022$ | $0.533 \pm 0.007$ | $0.336 \pm 0.004$ | $0.629 \pm 0.003$ | $0.559 \pm 0.027$ | $0.544 \pm 0.027$ | $0.247 \pm 0.003$ | $0.353 \pm 0.004$ | $0.627 \pm 0.004$ |
| | TSEF (w/o $\mathbf{M}_f$) | $0.660 \pm 0.038$ | $0.683 \pm 0.032$ | $0.411 \pm 0.006$ | $0.250 \pm 0.004$ | $0.602 \pm 0.003$ | $0.672 \pm 0.030$ | $0.660 \pm 0.033$ | $0.253 \pm 0.003$ | $0.368 \pm 0.004$ | $0.626 \pm 0.004$ |
| | **TSEF (FULL)** | $\mathbf{0.843 \pm 0.030}$ | $\mathbf{0.852 \pm 0.027}$ | $\mathbf{0.545 \pm 0.007}$ | $\mathbf{0.341 \pm 0.004}$ | $\mathbf{0.669 \pm 0.003}$ | $\mathbf{0.789 \pm 0.012}$ | $\mathbf{0.781 \pm 0.014}$ | $\mathbf{0.334 \pm 0.004}$ | $\mathbf{0.376 \pm 0.004}$ | $\mathbf{0.638 \pm 0.004}$ |

as a practical *attack knob* that allows an adversary to navigate the stealth–success spectrum: smaller $\lambda_{\text{cls}}$ favors high-fidelity cover-up under strict auditing, whereas larger $\lambda_{\text{cls}}$ prioritizes brute-force misclassification when stealth is secondary.

## H.5. Ablation study

Table 9 ablates TSEF's temporal vulnerability mask $\mathbf{M}_t$ (**w/o $\mathbf{M}_t$**: global spectral update) and frequency perturbation filter $\mathbf{M}_f$ (**w/o $\mathbf{M}_f$**: localized PGD update). **TSEF (Full)** yields the best joint trade-off, combining strong targeting (F1/ASR) with higher explanation alignment (AUPRC/AUP/AUR).

Removing $\mathbf{M}_t$ causes the largest drop in targeting, especially on SEQCOMB-MV: for TIMEX++, ASR $0.789 \rightarrow 0.491$ and F1 $0.795 \rightarrow 0.508$; for INTEGRATED GRADIENTS, ASR $0.781 \rightarrow 0.544$. This indicates temporal localization is necessary to concentrate the budget on vulnerable windows and reach the target label.

Removing $\mathbf{M}_f$ also degrades the trade-off: on SEQCOMB-MV+TIMEX++, ASR $0.789 \rightarrow 0.678$ and AUPRC $0.661 \rightarrow 0.632$; on LOWVAR+TIMEX++, ASR $0.848 \rightarrow 0.657$. For INTEGRATED GRADIENTS on SEQCOMB-MV, enabling $\mathbf{M}_f$ improves explanation alignment (AUPRC $0.253 \rightarrow 0.334$, AUP $0.368 \rightarrow 0.376$). Together, $\mathbf{M}_t$ is key for effective targeting, while $\mathbf{M}_f$ improves coherent explanation control, and both are needed to satisfy TSEF's dual objectives.

