# OpenReview forum: "Exposing Vulnerabilities in Explanation for Time Series Classifiers via Dual-Target Attacks"
_ICML.cc/2026/Conference — ICML 2026 regular_

### Official Review · Reviewer_MpCz · 2026-03-01

**Soundness:** 2
**Presentation:** 3
**Significance:** 3
**Originality:** 3
**Overall Recommendation:** 4
**Confidence:** 3

**Summary:**

This paper studies the robustness of interpretable time-series deep learning systems under adversarial perturbations. The main claim is that explanation stability is not a reliable proxy for decision robustness: an attacker can simultaneously force a target misclassification while keeping the explanation aligned with a chosen “reference rationale”. To demonstrate this vulnerability, the authors introduce TSEF, a dual-target white-box attack that jointly optimizes a Temporal Vulnerability Mask (TVM) to localize sensitive regions and a Frequency Perturbation Filter (FPF) to apply structured spectral modifications within those regions.

**Compliance With Llm Reviewing Policy:**

Affirmed.

**Final Justification:**

I have updated my score after the rebuttal

**Key Questions For Authors:**

1. How well does TSEF transfer across different classifier backbones (e.g., CNN/TCN vs transformer)?

2. How whould practitioners interpret the risk in black-box settings?

**Limitations:**

No — the paper should more explicitly discuss the gap between white-box evaluation and practical attacker capabilities.

**Strengths And Weaknesses:**

**Soundness**:

*Strengths:*
The paper clearly formalizes the dual-target objective (target label + target explanation) and proposes a method that directly operationalizes the theoretical insight (localize edits + time-frequency perturbations).

*Weakness:*
1. The evaluation is entirely white-box. This is a reasonable first step, but it limits conclusions about real-world feasibility. It would strengthen the work to evaluate a black-box variant.

2. Classifiers are transformer-based; robustness across other common architectures is not evaluated.

**Presentation:**
Clear structure and motivation-to-method alignment.

**Significance:**
The paper exposes a practically important failure mode for explanation-based auditing of time-series models and motivates a more appropriate coupled robustness evaluation.

**Originality**
The dual-target framing and the time-series-specific structured time–frequency attack design constitute a meaningful contribution.

---

> ### Author Rebuttal · Authors · 2026-03-31
>
> > W1: White-box setting
>
> * Thank you for the suggestion. We agree that the current study is limited to the white-box setting, and we will make this scope more explicit in the revision.
> * We adopt white-box evaluation as a **worst-case robustness stress test** of the coupled predictor–explainer system, following standard adaptive robustness evaluation [1–6].
> * Extending TSEF to a black-box setting is non-trivial because the current attack optimizes a joint prediction–explanation objective and therefore requires predictor/explainer gradients.
> * To partially address this concern, we also conduct a preliminary **black-box proxy** (please see the table in our response to Q1) via surrogate transfer (Transformer + TimeX++ $\rightarrow$ CNN + TimeX++). Transfer is weaker than white-box but non-zero, and TSEF remains slightly stronger than ADV$^2$.
> * We will clarify that the current paper is primarily a white-box stress test, while full black-box joint attacks remain an important future direction.
>
> [1] Goodfellow et al. Explaining and harnessing adversarial examples. ICLR 2015. [2] Madry et al. Towards deep learning models resistant to adversarial attacks. ICLR 2018. [3] Athalye et al. Obfuscated gradients give a false sense of security. ICML 2018. [4] Tramer et al. On adaptive attacks to adversarial example defenses. NeurIPS 2020. [5] Ghorbani et al. Interpretation of neural networks is fragile. AAAI 2019. [6] Croce and Hein. Reliable Evaluation of Adversarial Robustness with an Ensemble of Diverse Parameter-Free Attacks. ICML 2020.
>
> > W2: Robustness Beyond Transformers
>
> We thank the reviewer for this suggestion. To address it, we additionally evaluate TSEF on a CNN-based classifier and a TimesFM 2.5-based classifier, both paired with TimeX++. The new results support the same conclusion as in the main paper: the vulnerability is not limited to the original transformer backbone. On SEQCOMB-UV with CNN, TSEF remains competitive on prediction while improving ASR and all explanation metrics. On SEQCOMB-UV with TimesFM 2.5, TSEF outperforms both PGD and ADV$^2$ on all five metrics. Full tables for additional datasets are available at: https://anonymous.4open.science/r/TSEF-Rebuttal-Results-3CFA. We will include these results in the revision.
>
> **TimesFM 2.5 Attack Results (Higher is better):**
>
> | Dataset | Method | F1 | ASR | AUPRC | AUP | AUR |
> |---|---|---|---|---|---|---|
> | SEQCOMB-UV | PGD | 0.6615±0.0319 | 0.6620±0.0357 | 0.2731±0.0028 | 0.2718±0.0028 | 0.6460±0.0031 |
> | | ADV$^2$ | 0.6169±0.0260 | 0.6243±0.0324 | 0.2834±0.0030 | 0.2798±0.0029 | 0.6471±0.0031 |
> | | TSEF | **0.7900±0.0403** | **0.7870±0.0414** | **0.3258±0.0033** | **0.3218±0.0032** | **0.6520±0.0031** |
>
> **CNN Attack Results (Higher is better):**
> | Dataset | Method | F1 | ASR | AUPRC | AUP | AUR |
> |---|---|---|---|---|---|---|
> | SEQCOMB-UV | PGD | 0.5572±0.0205 | 0.5301±0.0222 | 0.1565±0.0019 | 0.1332±0.0018 | 0.6441±0.0033 |
> | | ADV$^2$ | **0.5574±0.0201** | 0.5291±0.0227 | 0.1697±0.0020 | 0.1389±0.0018 | 0.6515±0.0033 |
> | | TSEF | 0.5560±0.0194 | **0.5324±0.0207** | **0.2033±0.0024** | **0.1611±0.0021** | **0.6649±0.0032** |
>
> > Q1: Attack Transferability
>
> Thank you for the question. To assess cross-backbone transferability, we generate adversarial examples on a Transformer + TimeX++ source pair and evaluate them on an unseen CNN + TimeX++ target pair. The results show non-zero transferability. Compared with ADV$^2$, TSEF achieves higher F1/ASR on all three datasets and improves most explanation metrics. This provides preliminary evidence that the coupled vulnerability exposed by TSEF is not limited to a single classifier backbone. We will include these results and clarify in the revision.
>
> **Transfer Attack Results (Higher is better):**
> | Dataset | Method | F1 | ASR | AUPRC | AUP | AUR |
> |---|---|---|---|---|---|---|
> | LOWVAR | ADV$^2$ | 0.2873±0.0464 | 0.2910±0.0474 | 0.3487±0.0052 | 0.2735±0.0049 | 0.3984±0.0041 |
> | | TSEF | **0.2887±0.0443** | **0.3014±0.0450** | **0.3608±0.0053** | **0.2805±0.0049** | **0.4051±0.0041** |
> | SEQCOMB-UV | ADV$^2$ | 0.0659±0.0059 | 0.0913±0.0123 | 0.1521±0.0020 | 0.1315±0.0019 | 0.6507±0.0034 |
> | | TSEF | **0.0719±0.0036** | **0.1021±0.0103** | **0.1532±0.0020** | **0.1322±0.0019** | **0.6513±0.0034** |
> | SEQCOMB-MV | ADV$^2$ | 0.0239±0.0062 | 0.0253±0.0069 | 0.5845±0.0052 | **0.6106±0.0052** | 0.5726±0.0037 |
> | | TSEF | **0.0369±0.0092** | **0.0395±0.0106** | **0.6256±0.0054** | 0.6082±0.0052 | **0.5960±0.0035** |
>
> > **Q2**: How whould practitioners interpret the risk in black-box settings?
>
> * In practice, our results should be interpreted as a worst-case warning for explanation-based auditing, not as evidence that every black-box deployment is equally vulnerable.
> * The key implication is that explanation stability alone is not a reliable robustness signal.
> * Explanation-based monitoring should be complemented with coupling-aware robustness checks.

---

> > ### Author Rebuttal · Reviewer_MpCz · 2026-04-05
> >
> > I have updated my score after the rebuttal

---

### Official Review · Reviewer_rTPP · 2026-03-03

**Soundness:** 2
**Presentation:** 2
**Significance:** 2
**Originality:** 3
**Overall Recommendation:** 4
**Confidence:** 3

**Summary:**

This paper proposes a novel adversarial attack approach. The paper first demonstrates the "high-dimensional paradox”, which reveals that although high-dimensional spaces provide a massive attack surface, applying unstructured perturbations makes it impossible to precisely control the resulting explanation. To address this, the paper proposes a structured attack that decouples the perturbation process into "when to edit" (via a temporal mask) and "how to edit" (via a frequency filter). The study conducted experiments on three synthetic and three real-world datasets, verifying that the proposed attack effectively misleads the classifier while maintaining a plausible and consistent explanation.

**Compliance With Llm Reviewing Policy:**

Affirmed.

**Final Justification:**

Thanks for the authors' response. I have updated the score accordingly.

**Key Questions For Authors:**

1. How is the reference explanation $A'$ practically derived or selected in a real-world scenario? Furthermore, why is the adversarial goal (in left bottom of page 3) formulated to steer toward a distinct $A'$, rather than simply maintaining the original explanation $A$ (which could arguably be stealthier)?

2. Why is the strict white-box assumption considered sufficient for your current threat model? What specific technical limitations prevented the exploration of a black-box setting?

3.  Given the vulnerabilities exposed by TSEF, what specific defense perspectives or mitigation strategies would you recommend? Have you evaluated whether any existing defense mechanisms can effectively detect or neutralize this attack?

**Limitations:**

A dedicated "Limitations" section is strongly recommended. Proactively acknowledging the difficulty of executing attacks under non-white-box conditions and the current gaps in defense research will not only significantly enhance the scientific rigor of the paper but also provide a highly valuable research roadmap for future defensive directions.

**Strengths And Weaknesses:**

## Soundness

* This work demonstrates the challenges of the "high-dimensional paradox”, and then logically decouples the attack into "when to edit" and "how to edit".

* The paper clearly explains the design rationale behind the TSEF framework.

* Furthermore, the experiments are comprehensively designed across both synthetic and real-world datasets.

However, the weakness on the soundness are as follows:

* Justification of the reference explanation and threat model.

The paper defines the adversarial objective in Section 3 as steering the manipulated explanation towards a designated reference explanation $A'$, rather than maintaining the original explanation $A$. However, the paper lacks a clear justification for how this reference $A'$ is practically selected in a real-world threat model. Intuitively, shifting the explanation from $A$ to a completely different $A'$ might introduce noticeable discrepancies, making the attack less stealthy and easier for human auditors or rule-based auditor to detect. Maintaining the original explanation ($A' = A$) while successfully altering the prediction, often known as an explanation invariance attack, could arguably offer higher stealth. The authors should clarify the practical motivation for targeting a specific $A'$ and discuss why an $A' = A$ objective was not considered or compared as a stealthier baseline.

* Lack of evaluation on frontier time series foundation models.

The current experimental setup evaluates the proposed method on traditional deep learning architectures, missing recent advancements in the field. To robustly demonstrate the soundness and generalizability, the authors should consider incorporating some SOTA time series foundation models into their evaluation. For instance, commercial models like TimeGPT (which supports anomaly detection) and open-source models like Google's TimesFM 2.5 (which can be easily adapted for time series classification) are highly relevant.

* Incomplete discussion on suboptimal performance.

In Tables 1 and 3, there are several specific metrics and experimental settings where baseline attacks outperform the proposed TSEF method. The paper currently lacks a sufficient and transparent analysis of these instances. To improve the soundness of the paper, the authors must explicitly acknowledge these results and provide an in-depth discussion detailing the specific conditions, datasets, or explainer backbones under which TSEF falls short compared to baselines, along with the underlying technical reasons.

## Presentation

The paper follows a standard academic structure, making it generally easy to follow. However, there are several presentation issues that need to be addressed:

* Figure 2 is hard to follow. Placing $y'$ and $A'$ in the right corner misleads the reader and obscures the overarching workflow of the TSEF attack. Therefore a more clearly TSEF overview is required to improve the presentation.
* Missing metric definitions. The paper fails to define the acronyms for the explanation metrics. What exactly do AUPRC, AUR, and AUP stand for?
* Terminology misuse. The term "Threat Models" is misused in some section headings (e.g., Section 5.1 and Section 5.2), as these sections seem to describe experimental setups rather than the "threat models".

## Significance

The authors propose a novel adversarial attack on time series data, motivated by the "high-dimensional paradox”. Compared to previous methods, this work achieves higher stealth because the perturbed data maintain a plausible explanation despite yielding an incorrect classification. However, the overall significance and practical impact of the work are currently limited by the following factors:

* Lack of justification for the white-box assumption. The paper focuses entirely on white-box attacks. The paper fails to discuss why this strong assumption (requiring full access to model parameters and gradients) is sufficient or realistic for their threat model. Furthermore, there is no discussion regarding the technical limitations that prevent the extension of the TSEF attack to a more realistic black-box setting.

* Missing evaluation of attack transferability. The paper lacks a critical discussion and evaluation of the attack's transferability. It is crucial to know whether the generated attacked time series using one predictor-explainer pair can successfully fool a different, unseen target predictor and explainer. Exploring whether the TSEF method can transfer across different architectures would significantly strengthen the practical relevance of the attack.

* Absence of discussion on mitigation strategies. While the paper proposes a potent attack method, it largely omits discussions on potential countermeasures(i.e., defense mechanism).

## Originality

The authors first introduce the high-dimensional paradox: Although high-dimensional space provides a massive attack surface, applying unstructured perturbations in these high dimensions makes it impossible to precisely control the explanation. To overcome this, the authors propose a structured attack by decoupling the manipulation into "when to edit" (via a temporal mask) and "how to edit" (via a frequency filter). Overall, I believe the originality of this work is good and there is no significant issue from my side on the originality.

---

> ### Author Rebuttal · Authors · 2026-03-31
>
> > W1: Foundation Models
>
> We agree that evaluation on frontier time-series foundation models would strengthen the paper. We additionally evaluate TSEF on a TimesFM 2.5-based classifier with TimeX++. The new results support our main claim that the prediction–explanation vulnerability is not limited to the original backbones. On SEQCOMB-UV, TSEF improves all five metrics over PGD and ADV$^2$. Full table for additional datasets is available at: https://anonymous.4open.science/r/TSEF-Rebuttal-Results-3CFA, and will be included in the revision. Due to rebuttal-time constraints, we include TimesFM 2.5 here and will work on additional foundation models.
>
> TimesFM Attack Results (Higher is better):
> | Data | Method | F1 | ASR | AUPRC | AUP | AUR |
> |---|---|---|---|---|---|---|
> | SEQCOMB-UV | PGD | 0.6615 | 0.6620 | 0.2731 | 0.2718 | 0.6460 |
> | | ADV$^2$ | 0.6169 | 0.6243 | 0.2834 | 0.2798 | 0.6471 |
> | | TSEF | **0.7900** | **0.7870** | **0.3258** | **0.3218** | **0.6520** |
>
> > W2: Suboptimal Cases
>
> We agree and will make these cases explicit. The suboptimal cases mainly occur on isolated prediction metrics (ASR/F1), especially in easy misclassification regimes such as ECG, where prediction-only attacks can devote the full budget to label flipping. However, they are much weaker on explanation alignment. Thus, the key result is joint effectiveness: TSEF remains competitive on prediction while being substantially stronger on explanation metrics, whereas existing baselines are typically strong on only one side.
> > W3: Figure 2 Clarity
>
> We will revise Figure 2 to clarify the TSEF workflow
> > W4: Explanation Metrics
>
> We will define AUPRC, AUP, and AUR explicitly in the main text; these are the same explanation-alignment metrics used in TimeX/TimeX++ papers, with details already provided in the appendix H.
> > W5/Q2: White-Box Assumption & Black-Box Limitations
>
> * Thank you for the question. We agree that the current study is limited to the white-box setting, and we will make this scope more explicit in the revision.
> * We adopt white-box evaluation as a **worst-case robustness stress test** of the coupled predictor–explainer system. This follows standard adversarial-robustness practice, where adaptive white-box attacks are used to assess worst-case vulnerability [1-6].
> * Extending TSEF to black-box settings is non-trivial because the attack optimizes a **joint prediction–explanation objective**, which in its current form requires access to predictor/explainer gradients.
> * As a preliminary **black-box proxy**, we also conducted a cross-backbone surrogate-transfer experiment (Transformer + TimeX++ $\rightarrow$ CNN + TimeX++). The transfer results (please see the table in our response to W6) show TSEF remains slightly stronger than ADV$^2$ on nearly all metrics across the reported datasets. We will add this evidence and clarify that the current paper primarily studies **white-box stress testing**, while black-box joint attacks remain an important future direction.
>
> Please see the full reference list [1-6] in our response to W1 of Reviewer MpCz.
>
> > W6: Attack Transfer
>
> We agree that transferability is important for practical relevance. We therefore add a cross-backbone transfer experiment: adversarial examples are generated on a Transformer + TimeX++ source pair and evaluated on an unseen CNN + TimeX++ target pair.
>
> Compared with ADV$^2$, TSEF achieves higher F1/ASR on all three datasets and improves most explanation metrics. Please see the anonymous link in our response to W1 for full results on additional datasets.
>
> Transfer Results (Higher is better):
> | Data | Method | F1 | ASR | AUPRC | AUP | AUR |
> |---|---|---|---|---|---|---|
> | SEQCOMB-UV | ADV$^2$ | 0.0659 | 0.0913 | 0.1521 | 0.1315 | 0.6507 |
> | | TSEF | **0.0719** | **0.1021** | **0.1532** | **0.1322** | **0.6513** |
> > W7/Q3: Mitigation & Defenses
>
> Our goal is to expose a previously under-studied vulnerability in interpretable time series systems, rather than to propose a defense. To our knowledge, this is the first stress test of coupled adversarial robustness, and we hope it motivates future coupling-aware defenses and audits. One mitigation direction is joint adversarial training or regularization under a coupled prediction–explanation objective.
> > Q1: Reference Explanation
>
> In our setup, $A^{\prime}$ is the ground-truth rationale when annotations are available, and otherwise is derived from the clean explanation of the same interpreter. The preserving the original explanation case is a special case of this formulation, since in our main auditing setting $A^{\prime}$ is the original-class rationale.
> > W8: Limitations Section
>
> We agree that a dedicated Limitations section would strengthen the paper. In the revision, we will explicitly discuss the current focus on white-box, differentiable settings and the added difficulty of black-box, and position this work as a first stress test of coupled robustness that motivates future coupling-aware defenses.

---

> > ### Author Rebuttal · Reviewer_rTPP · 2026-03-31
> >
> > 1. Defenses and Black-Box Extensions are acknowledged but haven't been fleshed out.
> >
> > 2. The requested comparison against an $A^{′} = A$ stealth baseline is not included, and a clearer justification of the practical threat model, especially the shift to a different $A^{′}$, would strengthen the work.
> >
> > 3. The additional experiment indicates the method is highly tailored to the white-box setting, with limited evidence of practical black-box applicability.

---

> > > ### Author Response · Authors · 2026-04-06
> > >
> > > > Q1 & Q3: Practical Black-Box Applicability and Defenses
> > >
> > > Thank you for the comment. We agree that defenses and fully black-box extensions are important, but they are beyond the main scope of this paper. Our contribution is a **stress test of worst-case coupled predictor–explainer robustness**; coupling-aware defenses and stronger black-box joint attacks are important future directions.
> > >
> > > To further probe applicability beyond the exact attacked model, we trained independent target Transformer+TimeX / Transformer+TimeX++ pairs on LowVar and evaluated surrogate transfer in both directions (TimeX++ → TimeX and TimeX → TimeX++). This approximates a more practical unseen-target setting in which the attacker does not have access to the exact target model/explainer. As shown below, TSEF again outperforms PGD and ADV$^2$ on all five metrics, improving both transfer targeting success (F1/ASR) and explanation alignment (AUPRC/AUP/AUR). Thus, the added results provide additional surrogate-transfer evidence that the vulnerability extends to practical black-box scenarios, while the paper’s main claim remains unchanged: targeted misclassification can coexist with preserved explanations under explanation-based auditing.
> > >
> > > | Transfer Pair | Method | F1 | ASR | AUPRC | AUP | AUR |
> > > |---|---|---|---|---|---|---|
> > > | TimeX++ $\rightarrow$ TimeX | PGD | 0.6287±0.0391 | 0.6550±0.0342 | 0.2179±0.0042 | 0.1481±0.0034 | 0.3545±0.0035 |
> > > | | ADV$^2$  | 0.6226±0.0457 | 0.6507±0.0395 | 0.2746±0.0045 | 0.1886±0.0036 | 0.4129±0.0036 |
> > > | | TSEF | **0.6514±0.0343** | **0.6686±0.0321** | **0.3606±0.0050** | **0.2477±0.0038** | **0.4976±0.0037** |
> > > | TimeX $\rightarrow$ TimeX++ | PGD | 0.6583±0.0343 | 0.6730±0.0301 | 0.2564±0.0043 | 0.1945±0.0040 | 0.2983±0.0036 |
> > > | | ADV$^2$  | 0.6530±0.0384 | 0.6684±0.0344 | 0.3385±0.0049 | 0.2559±0.0044 | 0.3710±0.0041 |
> > > | | TSEF | **0.6655±0.0317** | **0.6769±0.0285** | **0.4224±0.0054** | **0.3203±0.0047** | **0.4487±0.0045** |
> > >
> > > > Q2: $A^{\prime}=A$ Baseline and Justification of Threat Model
> > >
> > > Thank you for the clarification. In the practical annotated setting, our threat model is not a shift to an arbitrary different $A^{\prime}$; rather, $A^{\prime}$ is the auditor's ground-truth reference rationale for checking whether the model relies on the correct evidence. This setting is practically important because, in many high-stakes applications, explanations are used to verify not only whether the prediction is correct, but whether it is supported by the right evidence. An attack that preserves prediction plausibility while steering the explanation away from the auditor’s reference rationale can therefore directly undermine explanation-based auditing.
> > >
> > > The original-explanation-preserving case is a special case of this threat model. Following your suggestion, we additionally evaluate the $A^{\prime}=A$ setting, where $A$ is the original explanation. On the real-world datasets PAM and Epilepsy, which lack human annotated explanations, TSEF outperforms PGD and ADV$^2$ on all five metrics, with the same trend also observed on LowVar. Thus, our main conclusion remains unchanged: even in the stealthier original-explanation-preserving setting, targeted prediction manipulation can coexist with explanation preservation, and TSEF remains consistently stronger than prior baselines.
> > >
> > >
> > > | Dataset | Method | F1 | ASR | AUPRC | AUP | AUR |
> > > |---|---|---|---|---|---|---|
> > > | LowVar | PGD | 0.8441±0.0400 | 0.8568±0.0355 | 0.2536±0.0043 | 0.1915±0.0040 | 0.3185±0.0037 |
> > > | | ADV$^2$ | 0.8465±0.0396 | 0.8578±0.0355 | 0.6245±0.0052 | 0.4936±0.0046 | 0.6233±0.0040 |
> > > | | TSEF | **0.8703±0.0435** | **0.8839±0.0380** | **0.8342±0.0035** | **0.6938±0.0034** | **0.7700±0.0025** |
> > > | PAM | PGD | 0.6041±0.0124 | 0.5967±0.0142 | 0.6874±0.0057 | 0.6390±0.0036 | 0.4443±0.0040 |
> > > | | ADV$^2$ | 0.6091±0.0170 | 0.5967±0.0182 | 0.7542±0.0050 | 0.6719±0.0033 | 0.4519±0.0039 |
> > > | | TSEF | **0.6168±0.0182** | **0.6054±0.0185** | **0.8503±0.0037** | **0.6981±0.0031** | **0.4637±0.0038** |
> > > | Epilepsy | PGD | 0.1595±0.0111 | 0.1908±0.0158 | 0.6582±0.0061 | 0.3454±0.0044 | 0.8640±0.0021 |
> > > | | ADV$^2$ | 0.1595±0.0111 | 0.1908±0.0158 | 0.6783±0.0059 | 0.3468±0.0043 | 0.8670±0.0021 |
> > > | | TSEF | **0.1610±0.0149** | **0.1937±0.0212** | **0.7661±0.0046** | **0.3549±0.0042** | **0.8895±0.0021** |

---

### Official Review · Reviewer_PE5p · 2026-03-04

**Soundness:** 3
**Presentation:** 3
**Significance:** 3
**Originality:** 3
**Overall Recommendation:** 4
**Confidence:** 2

**Summary:**

This paper challenges the assumption that temporal consistency in explanations indicates robustness in interpretable time series systems, demonstrating that predictions and explanations can be adversarially decoupled. The authors introduce TSEF, a dual-target attack that achieves targeted misclassification while maintaining plausible explanations, revealing that explanation stability is a misleading proxy for decision robustness. Consequently, they argue for coupling-aware robustness evaluations to ensure trustworthy time series tasks.

**Compliance With Llm Reviewing Policy:**

Affirmed.

**Final Justification:**

This paper has good writing and comprehensive experiments. I will keep the positive score.

**Key Questions For Authors:**

1.  Why do we need a frequency perturbation filter and an energy-invariant?
  2.  What is the difference between a vision language adversarial attack and a time series?

**Limitations:**

The authors do not provide the limitations. The attack is based on the explanations and predictions; the authors do not consider the case that the explanations are not available.

**Strengths And Weaknesses:**

strength:
  1.  This paper proposes an idea that the prediction and explanation can be decoupled. This paper uses an attack method to verify this idea. Once the attack only modifies the explanation part, then only the prediction would change; the explanation remains the same.
  2.  This paper provides extensive experimental results. The ASR results reflect that the attack method is effective.
  3. This paper is well-organized, and the overall structure of the manuscript makes the proposed approach and experimental evaluation relatively easy to follow.

weakness:
  1.  Section 4.1 is confusing. Some terms are unclear, such as the $sign(g_c)$. In addition, $l_{\infty}$ is unclear.  This title, “High-Dimensional Paradox,” is confusing. The section describes the attack success ratio in prediction and explanations. It is not very related to high dimensions. It is not clear what the purpose of the theorem is.
  2.  The metric needs verifications for F1. Should the F1 results for the attack be lower, the better?

---

> ### Author Rebuttal · Authors · 2026-03-31
>
> > W1: Sec. 4.1 notation & High-Dimensional Paradox
>
> Thank you for pointing this out. We will clarify the notation in the revision: $g_c$ is the classification gradient, $\operatorname{sign}\left(g_c\right)$ denotes its element-wise sign, and the $\ell_{\infty}$ constraint means the maximum absolute change at any time-channel entry is bounded by $\epsilon$.
>
> More importantly, the purpose of Sec. 4.1 is not to explain ASR, but to formalize why dense $\ell_{\infty}$ attacks over **the full high-dimensional input** are poorly matched to the goal of targeted prediction plus localized explanation control. Theorem 4.1 shows that a one-step dense update increases attribution mass outside the target region $\Omega$, **so the mismatch to the reference explanation grows with dimensions**. The "High-Dimensional Paradox" is that **high dimensionality** helps dense attacks flip the label, but also makes it harder to keep the explanation localized and contiguous. This is exactly why TSEF uses a structured design: $M_t$ controls when to perturb, and $M_f$ controls how to perturb in a pattern-preserving way. We will revise Sec. 4.1 and its title to make this motivation clearer.
>
> > W2: F1 metric interpretations
>
> Thank you for pointing this out. Here, higher F1 is better. Our F1 is the **target-class F1** in the targeted attack setting, not the clean-label F1: for each attacked sample, we treat the assigned target label $y^{\prime}$ as the reference label and compute F1 between the adversarial prediction and $y^{\prime}$. Thus, ASR measures whether the attack reaches the target label, while target-class F1 measures how consistently the adversarial predictions match the desired target labels. We will make this explicit in the main text; Additional details are provided in Appendix H.2.
>
> > W3: The case that the explanations are not available.
>
> Thank you for the comment. We address this case in the paper. In Sec. 5.1, when ground-truth explanations are unavailable, we use the top-$k \%$ salient region from the clean explanation of the same interpreter as the reference explanation. We also evaluate datasets without human-annotated rationales and discuss real-world deployments where the clean explanation serves as the auditing reference. We will clarify this more explicitly in the revision.
>
> > Q1: Frequency filter & energy invariance
>
> Thank you for the question. We need $M_f$ because the attack must not only change the label, but also produce a plausible, contiguous explanation. Dense time-domain updates can flip predictions, but often fail to match a target rationale. TSEF therefore decouples when to attack and how to attack: $M_t$ localizes a vulnerable region, and $M_f$ edits it in the frequency domain to induce coherent changes in trends/periodicities rather than scattered point-wise noise.
>
> The energy-invariant update is needed because ordinary frequency-domain gradients are biased toward high-energy bins. Using a sign step removes this scale bias, so the update depends on alignment with the joint attack objective rather than raw spectral energy. This is also supported by ablations in Table 2 of our paper: removing $M_f$ weakens the joint trade-off between targeted prediction and explanation alignment. Thus, $M_f$ is not just an implementation detail; it enables coherent spectral edits that better satisfy the dual objective.
>
> > Q2: Differences from vision/NLP attacks
>
> Thank you for the question. As we state in the introduction, our setting is "fundamentally different" from joint prediction/explanation attacks in vision/NLP for two time-series-specific reasons: (i) pattern-level control is essential, since time series models respond to temporal structures such as trends and periodic components, so naive per-timestep perturbations may flip the label but need not produce the desired explanation; and (ii) time series exhibit a high dimensional paradox, where unconstrained optimization over the $T \times D$ input tends to create dense, temporally incoherent updates that are hard to steer toward a reference rationale. We also note this explicitly in Sec. 4.1, where dense $\ell_{\infty}$ attacks are shown to disperse attribution outside the target region, yielding diffuse rather than localized, contiguous explanations.
>
> This is exactly why TSEF uses a time-frequency structured design instead of directly porting vision style joint attacks. As discussed in Sec. 4.2, TSEF decouples the problem into when to attack and how to attack. By contrast, a vision baseline such as ADV$^2$ optimizes a dense joint objective; in our time series setting, such unconstrained updates are much more likely to scatter attribution mass globally rather than synthesize the pattern-like rationales expected in time series auditing. Thus, the key difference is not just the input modality, but the need to preserve the temporal topology of the explanation while changing the prediction.

---

> > ### Author Rebuttal · Reviewer_PE5p · 2026-04-01
> >
> > Thanks for the authors' response. I do not have any questions.

---

### Official Review · Reviewer_mQUm · 2026-03-05

**Soundness:** 3
**Presentation:** 3
**Significance:** 4
**Originality:** 3
**Overall Recommendation:** 5
**Confidence:** 4

**Summary:**

This paper aims to address the black box nature of deep neural networks. The general challenge is that, due to the black box nature of neural networks, it is difficult to determine why a particular model achieved high accuracy or why specific decisions or predictions were made. The more specific problem is that domain experts would pair a predictor with an explainer that can highlight the rationale behind decisions or predictions; however, this method primarily depends on the assumption that the model is faithful enough to be trusted by default. This can undermine human oversight and even downstream decision-making, especially if the explanation is erroneous. Moreover, there is a tendency for adversarial attacks to introduce unexpected perturbations, leading to changes in prediction. So these authors aim to answer the question “ Can we trust the joint prediction - explanation output of an interpretable time series deep learning system under adversarial perturbations?”

They try to address this question by introducing what they termed the Time Series Explanation Fooler (TSEF). This is an attack framework that models an adversarial sample to force a frozen classifier to a target class and steer the coupled explainer towards a reference explanation, basically corrupting the system.

**Compliance With Llm Reviewing Policy:**

Affirmed.

**Final Justification:**

I maintain my Accept recommendation. The paper introduces the Time Series Explanation Fooler (TSEF), an adversarial framework that jointly targets both predictions and explanations in interpretable time-series models. The problem is important, and the paper provides a technically sound formulation, detailed threat model, and comprehensive experiments across synthetic and real datasets. The evaluation and ablation studies support the effectiveness of the proposed attack and highlight vulnerabilities in coupled predictor–explainer systems.

My main concerns were related to the choice of explainers, lack of explicit limitations, and the need for clearer visualization of attack effects. The authors clarified that the selected explainers are differentiable and compatible with the white-box gradient-based formulation, while alternatives such as SHAP-based methods would require different optimization paradigms. They also provided preliminary transfer-attack results demonstrating partial generalization beyond the white-box setting and agreed to expand the limitations discussion. Additionally, they committed to adding a three-way visualization (original vs. attacked vs. defended), which will improve clarity.

Overall, the rebuttal adequately addressed my concerns and strengthened confidence in the scope and evaluation. I therefore maintain my Accept recommendation.

**Key Questions For Authors:**

1. Why were the selected explainers used (TimeX, TimeX++ and Integrated gradients)? There are other potential explainers that can be utilised, including Rocket-LIME, TsSHAP or SHAPEX. What was the rationale behind the selections made?

2. Are there any challenges or limitations to adopting your proposed methods ?

3. The authors should include a comparative visualization figure that demonstrates the explanation heatmaps across three distinct states (1) baseline/original, (2) under attack (without defense), and (3) under attack (with the proposed defense)

**Limitations:**

I think readers will benefit from understanding the limitations of this work. Include a discussion on the limitations and future directions of this study.

**Strengths And Weaknesses:**

This paper is fairly technically sound, and many of the claims have been adequately referenced. The preliminaries section does a good job of defining the time series classifier, providing explanations, and outlining the threat model and adversarial objectives. I would recommend a thorough review of the notations and equations in this section, and ensuring consistency across all notations.
The high-dimensional paradox challenge in attacking time-series classifiers and explainers was adequately introduced. The proposed approach was also well-detailed. The ablations and experimental approach were also well-detailed and informative. The use of several datasets, both synthetic and real, was also adequate to validate the proposed methods. The reference section is in excellent shape. All sources are properly cited, up-to-date, and follow the mandatory formatting template without deviation

---

> ### Author Rebuttal · Authors · 2026-03-31
>
> > Q1: Explainer Selection
>
> Thank you for the question. First, we selected TimeX, TimeX++, and Integrated Gradients because TSEF is formulated as a **white-box, gradient-based robustness stress test** of the coupled predictor–explainer system. This follows the standard adversarial-robustness protocol of evaluating models under a strongest adaptive white-box threat model, which has long been used to assess **worst-case robustness** in prior work [1-6]. White-box evaluation is therefore important here not merely to demonstrate an attack, but to test whether explanation stability remains reliable when an attacker can directly optimize against the full predictor–explainer pipeline.
>
> Second, under this formulation, the explainer must be directly compatible with gradient-based optimization with respect to the input. IG, TimeX, and TimeX++ satisfy this requirement, while also covering both a standard attribution method (IG) and recent time-series-specific neural explainers (TimeX/TimeX++).
>
> Third, alternatives such as Rocket-LIME, TsSHAP, and SHAPEX are not directly compatible with our current formulation in their standard form, since they rely on perturbation/sampling-based, surrogate-based, or discrete explanation pipelines rather than end-to-end differentiable mappings from input to explanation. As a result, attacking them would require different paradigms, such as query-based or black-box optimization, rather than our current white-box gradient-based objective.
>
> We agree that extending TSEF to such non-differentiable explainers is an important future direction, and we will clarify this motivation in the revision.
>
> [1] Goodfellow et al. Explaining and harnessing adversarial examples. ICLR 2015. [2] Madry et al. Towards deep learning models resistant to adversarial attacks. ICLR 2018. [3] Athalye et al. Obfuscated gradients give a false sense of security. ICML 2018. [4] Tramer et al. On adaptive attacks to adversarial example defenses. NeurIPS 2020. [5] Ghorbani et al. Interpretation of neural networks is fragile. AAAI 2019. [6] Croce and Hein. Reliable Evaluation of Adversarial Robustness with an Ensemble of Diverse Parameter-Free Attacks. ICML 2020.
> > Q2: Method limitations and challenges
>
> Yes. The main limitation of our current study is that TSEF is formulated in a white-box setting, requiring access to gradients of both the classifier and the explainer. As a result, the current formulation is limited to explainers that are directly compatible with gradient-based optimization.
>
> Another challenge is extending TSEF to black-box or non-differentiable explainers. A natural path is to replace the current gradient-based optimization with either surrogate-based transfer attacks (i.e., optimizing TSEF on a surrogate predictor–explainer pair and transferring to an unseen target pair) or query-based / zeroth-order optimization over the joint prediction–explanation objective.
>
> We also conducted a preliminary cross-backbone surrogate-transfer experiment (Transformer + TimeX++ $\rightarrow$ CNN + TimeX++) as **a black-box proxy**. The results (in the table below) show non-zero but weaker transferability than in the white-box setting, while TSEF still slightly outperforms ADV$^2$, a representative joint attack originally designed for image classifiers and explainers, on nearly all metrics across all three datasets. This provides preliminary evidence that the coupled vulnerability can transfer beyond the attacked backbone, while also confirming that the current paper remains primarily a white-box stress test of predictor–explainer vulnerability.
>
> We agree that these limitations should be stated more explicitly, and we will clarify them in the revision.
>
> Transfer Attack Results (Higher is better):
> | Dataset | Method | F1 | ASR | AUPRC | AUP | AUR |
> |---|---|---|---|---|---|---|
> | LOWVAR | ADV$^2$ | 0.2873±0.0464 | 0.2910±0.0474 | 0.3487±0.0052 | 0.2735±0.0049 | 0.3984±0.0041 |
> | | TSEF | **0.2887±0.0443** | **0.3014±0.0450** | **0.3608±0.0053** | **0.2805±0.0049** | **0.4051±0.0041** |
> | SEQCOMB-UV | ADV$^2$ | 0.0659±0.0059 | 0.0913±0.0123 | 0.1521±0.0020 | 0.1315±0.0019 | 0.6507±0.0034 |
> | | TSEF | **0.0719±0.0036** | **0.1021±0.0103** | **0.1532±0.0020** | **0.1322±0.0019** | **0.6513±0.0034** |
> | SEQCOMB-MV | ADV$^2$ | 0.0239±0.0062 | 0.0253±0.0069 | 0.5845±0.0052 | **0.6106±0.0052** | 0.5726±0.0037 |
> | | TSEF | **0.0369±0.0092** | **0.0395±0.0106** | **0.6256±0.0054** | 0.6082±0.0052 | **0.5960±0.0035** |
> > Q3: Visualization Request
>
> Thank you for the helpful suggestion. We currently provide Figure 3 to compare explanation heatmaps across attack methods. In the revision, we will add the suggested three-way figure showing the original input, the attacked input, and the attacked input after a defense/mitigation step. More broadly, our goal is to expose this vulnerability in interpretable time series systems and encourage future work on coupling-aware robustness evaluation and defenses.

---

> > ### Author Rebuttal · Reviewer_mQUm · 2026-04-01
> >
> > 1. The distinction between differentiable mappings and perturbation-based explainers like Rocket-LIME or SHAP justifies the current scope of the TSEF framework.
> > 2. The preliminary transfer-attack results provided in the rebuttal are helpful; they demonstrate that while the vulnerability is most potent in white-box settings, it does persist across backbones. I recommend including this table in the appendix of the final manuscript to provide a more comprehensive view of the attack's reach.
> > 3. Regarding the visualization, while I acknowledge Figure 3 in the current draft, the proposed 'three-way' figure (original vs. attacked vs. defended) is crucial for demonstrating the practical utility of any mitigation strategies. I look forward to seeing this and the promised notation cleanup in the camera-ready version.
> >
> >  I maintain my recommendation.

---

### Decision · Program_Chairs · 2026-04-30

**Decision:**

Accept (regular)

**Comment:**

Summary. This paper investigates the robustness of interpretable time-series deep learning systems under adversarial perturbations. The central claim is that explanation stability does not reliably indicate decision robustness: an attacker can simultaneously induce a target misclassification while keeping the explanation aligned with a chosen reference rationale. To demonstrate this vulnerability, the authors propose TSEF, a dual-target white-box attack that jointly optimizes a Temporal Vulnerability Mask (TVM) for localizing sensitive regions and a Frequency Perturbation Filter (FPF) for applying structured spectral modifications within those regions.

Reviewers' consensus. All reviewers lean toward acceptance (3 weak accept, 1 accept). Some reviewers mentioned that limitations of the work should be discussed, and the initially pointed to some potential concerns. The overall discussion addressed the majority of concerns raised by the reviewers.

Assessment. I concur with the reviewers: it represent an interesting and well-motivated work, that is important in the context of time series classifiers.